# Right Answer at the Right Time — Temporal Retrieval-Augmented Generation via Graph Summarization

**Zulun Zhu, Haoyu Liu, Mengke He, Siqiang Luo**
Nanyang Technological University

## Abstract

Question answering in temporal knowledge graphs requires retrieval that is both time-consistent and efficient. Existing RAG methods are largely semantic and typically neglect explicit temporal constraints, which leads to time-inconsistent answers and inflated token usage. We propose STAR-RAG, a temporal GraphRAG framework that relies on two key ideas: building a time-aligned rule graph and conducting propagation on this graph to narrow the search space and prioritize semantically relevant, time-consistent evidence. This design enforces temporal proximity during retrieval, reduces the candidate set of retrieval results, and lowers token consumption without sacrificing accuracy. Compared with existing temporal RAG approaches, STAR-RAG eliminates the need for heavy model training and fine-tuning, thereby reducing computational cost and significantly simplifying deployment. Extensive experiments on real-world temporal KG datasets show that our method achieves improved answer accuracy while consuming fewer tokens than strong GraphRAG baselines.

## 1 Introduction

Retrieval-augmented generation (RAG) has emerged as a practical remedy for the hallucination tendencies of large language models (LLMs) by grounding generation in external evidence, thereby substantially improving factuality across tasks such as question answering (Gutiérrez et al., 2024), text summarization (Edge et al., 2024), and decision support (Jiang et al., 2024). However, standard RAG remains largely document-centric, treating each document as an independent unit. This unstructured representation limits the ability to capture complex relational patterns among entities and events, and constrains multi-hop reasoning that requires composing evidence along structured paths (Pan et al., 2023; Besta et al., 2024). To address these limitations, recent studies explore graph-based RAG (GraphRAG), which organizes knowledge as graphs and retrieves relevant information over neighborhoods (Sarmah et al., 2024), paths (Delile et al., 2024), and subgraphs (Hu et al., 2024). Building on this direction, numerous works further improve retrieval by incorporating graph-structured knowledge and multi-hop sampling, enabling more accurate and efficient RAG frameworks (Peng et al., 2024).

However, real-world knowledge is inherently temporal: entities evolve, relations shift, and events carry explicit timestamps. While vanilla GraphRAG achieves strong retrieval performance on *static* knowledge, it encounters fundamental challenges when faced with time-sensitive queries: (i) retrieval remains dominated by semantic similarity and often overlooks explicit temporal constraints, producing answers that appear plausible yet are inconsistent with the question's time requirements; (ii) when naively applied to temporal knowledge, typical GraphRAG methods (Guo et al., 2024; Wang et al., 2024; Sun et al., 2024) tend to retrieve broadly across time rather than restricting to temporally relevant evidence, failing to adapt search strategies to the evolving nature of information. As a result, the token budget grows and accuracy degrades because the generator must filter temporal noise to isolate the small set of time-aligned evidence. This leads to both reduced efficiency and sub-optimal overall performance.

In response, several works systematically model the temporal structure of knowledge graphs from multiple perspectives. For example, Yang et al. (2025) improves time series forecasting by training a retrieval module that indexes historical patterns and feeding the retrieved subsequences into an LLM

for prediction. In a different vein, Wu et al. (2024a) adopts contrastive learning that compares queries with anchor events, training a shared encoder for event and query representations. Despite these insights, existing approaches frequently deviate from the original spirit of RAG, where the LLM is kept frozen and task-specific knowledge is supplied primarily through retrieval (Lewis et al., 2020; Gao et al., 2023). Instead, they often introduce heavy training pipelines that require parameter-dense encoders or fine-tuning the LLM itself across domains and time windows. This reliance on repeated optimization imposes considerable computational cost and hinders practical deployment. Thus, considerable effort is required when considering a more resource-friendly and efficient GraphRAG framework for temporal data.

In this paper, we propose STAR-RAG [1], an efficient RAG framework for temporal knowledge graphs that avoids any training or fine-tuning. STAR-RAG aims to simplify the retrieval process of RAG by aligning evidence selection with the question's temporal constraints, reducing both token and computation overhead. For this purpose, we first construct a rule graph that summarizes recurring categories of events as nodes, and links them with time-sensitive edges determined by how strongly one category tends to precede or follow another. The structure compresses individual events into a more compact form that preserves the key relational patterns for inference. Second, given a query, we identify a small set of seed events and run personalized PageRank (PPR) on the rule graph to prioritize their time-consistent neighborhood. By restricting graph propagation to the neighborhood around the seed nodes, we generate a concise candidate set that greatly reduces the search space yet preserves the most reliable evidence. Compared with the state-of-the-art MedicalGraphRAG (Wu et al., 2024b), our method improves answer accuracy by $9.1\%$ while reducing token usage by $97.0\%$. Our contributions are summarized as follows:

• We identify the limitations of existing GraphRAG on temporal data and propose STAR-RAG, a framework designed to achieve both high accuracy and efficiency in temporal question answering.

• We adopt two fundamental techniques to incorporate the question's temporal constraints and prioritize time-aligned neighborhoods: (i) constructing a rule graph that summarizes event categories and encodes temporal relations, and (ii) applying seeded personalized PageRank to focus retrieval on a time-aligned subgraph, thereby reducing the search space.

• We conduct comprehensive experiments on diverse real-world temporal knowledge graphs, demonstrating that STAR-RAG delivers higher answer accuracy while incurring fewer tokens than strong GraphRAG baselines.

## 2 RELATED WORKS

**Graph-Based Retrieval-Augmented Generation.** Text-based retrieval-augmented generation retrieves passages by semantic similarity between a question and texts, but it struggles to capture complex relational structure and topological patterns. To address these challenges, GraphRAG represents knowledge as nodes and edges and retrieves paths or subgraphs for structured reasoning. DALK (Li et al., 2024) constructs two domain-specific knowledge graphs from scientific corpora and applies coarse-to-fine sampling of knowledge graphs to select evidence fed back to LLMs. GRAG (Hu et al., 2024) retrieves $K$-hop ego-graphs scored by cosine similarity between query and textual embeddings, and then uses a learnable pruner to mask irrelevant nodes and edges before merging the top graphs into an optimal subgraph. SURGE (Kang et al., 2022) casts edges as nodes in a dual hypergraph and applies message passing with standard Graph Neural Networks (GNNs), which are trained contrastively to enforce knowledge-faithful responses. KGP (Wang et al., 2024) builds a passage-similarity knowledge graph while using an LLM to traverse it. To this end, it compiles a structured prompt for multi-document question answering. However, these static approaches remain time-agnostic, often overlooking explicit temporal constraints and producing answers that are semantically plausible yet temporally incorrect.

**Graph Summarization with MDL.** The graph summarization with minimum description length (MDL) optimization aims to find the optimal summary model which minimizes the bits to describe the model as well as the data given the model. Classic work (Navlakha et al., 2008) compresses a static graph into a coarse summary with corrections and explicitly combines this representation with MDL to yield intuitive and bounded-error summaries. For dynamic graphs, TimeCrunch (Shah et al.,

---

[1] Graph Summarization for Temporal Graph Retrieval-Augmented Generation

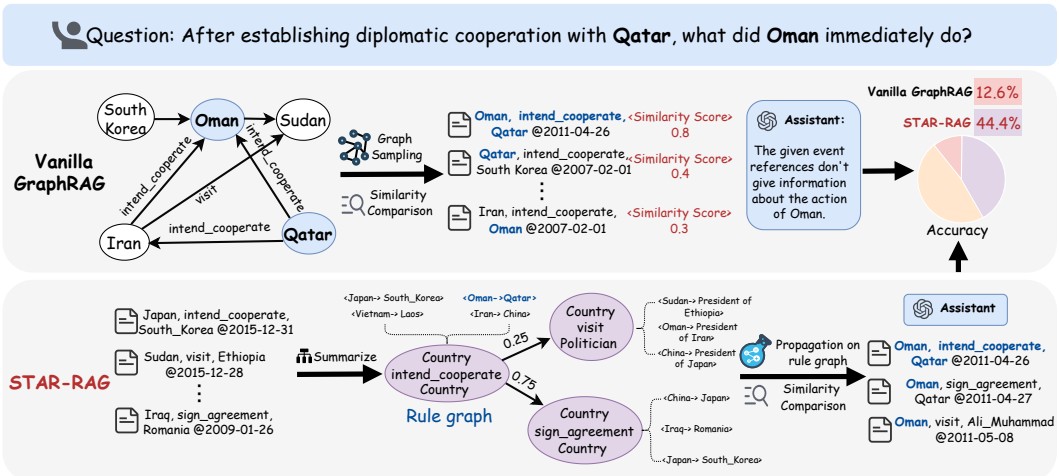

Figure 1: Comparison of vanilla GraphRAG and STAR-RAG. Vanilla GraphRAG relies primarily on semantic matching for retrieval and ignores explicit temporal constraints, which degrades answer accuracy. STAR-RAG maps events to a time-aligned rule graph and performs propagation on this graph to narrow the search space and prioritize time-consistent evidence, enabling temporally aligned retrieval and improved performance.

2015) formalizes summarization as minimizing the encoding cost of temporal structures and employs MDL-guided heuristics for model selection, achieving near-linear scalability on large real-world graphs. The most relevant to our work are KGist (Belth et al., 2020) and ANOT (Zhang et al., 2024a), which formulate knowledge graph summarization under the MDL principle and use it for effective anomaly detection. Unlike these methods, our MDL-guided summary builds a time-aligned rule graph that steers retrieval toward time-consistent evidence, thereby narrowing the search space at query time in GraphRAG.

**Reasoning on Temporal Knowledge Graphs.** Existing reasoning methods on temporal knowledge graphs often adopt a model-centric paradigm that trains specialized encoders or fine-tunes LLMs to capture temporal patterns. For instance, GenTKGQA (Gao et al., 2024) learns temporal GNN representations on retrieved subgraphs and instruction-tunes a language model to fuse these signals into generation. Similarly, GenTKG (Liao et al., 2023) couples temporal rule–based retrieval with specific instruction tuning for link prediction tasks. DyG-RAG Sun et al. (2025) builds an event-centric dynamic graph from text and performs time-aware event sequence retrieval with an LLM for temporal question answering. TimeR4 Qian et al. (2024) uses a temporal knowledge graph and a time aware retriever to rewrite questions, retrieve time consistent evidence, and then generate the final answer. However, such approaches typically depend on pairwise similarity over the full event set or require substantial training and maintenance, leading to low efficiency and limited scalability on large graphs.

## 3 METHODOLOGY

### 3.1 OVERVIEW

**Problem Definition.** A temporal knowledge graph $\mathcal{G}$ consists of temporal events in the form of $(s, r, o, t) \in \mathcal{F}$, where $s, o \in \mathcal{E}$, $r \in \mathcal{R}$ and $t \in \mathcal{T}$. Here $\mathcal{F}, \mathcal{E}, \mathcal{R}, \mathcal{T}$ denote the set of events, entities, relations, and valid timestamps in $\mathcal{G}$ respectively. Given a query $q$ in the set of questions, our objective is to efficiently retrieve the relevant information from $\mathcal{G}$ and obtain the answer extracted from the temporal events. The answer should be in a natural language form without violating the time constraint in $\mathcal{G}$.

In this section, we introduce the skeleton of our proposed method STAR-RAG, whose pipeline and comparison with vanilla GraphRAG are shown in Fig. 1. Unlike vanilla GraphRAG, which retrieves directly over the dense temporal knowledge graph, STAR-RAG first summarizes it into a much

---

**Algorithm 1:** STAR-RAG Retrieval

---

**Input:** query $q$; event set $\mathcal{F}$, $K_1$; $K_2$; restart $\alpha \in (0, 1)$

**Output:** Top-$K_1$ events corresponding to query $q$

1   $\mathcal{C}_s \leftarrow \text{ENTITYLABELING}(s)$ for each $s \in \mathcal{E}$      ▷ Apply Alg. 2 (in Appendix A) to assign the labels for each entity

2   $\mathcal{U} = \{\langle c_s, r, c_o \rangle | \exists\, (s, r, o, t) \in \mathcal{F}$ and $c_s \in \mathcal{C}(s), c_o \in \mathcal{C}(o)\}$      ▷ Collect the mapped rules as the candidate nodes

3   $\mathcal{W} = \{\{u, v\} : u \neq v, d_H(u, v) \leq 1\}$      ▷ Collect the candidate edges considering the Hamming distance of linked nodes

4   $\mathcal{W}' \leftarrow \emptyset$, $M \leftarrow \emptyset$

5   **while** $\exists\, (u, v) \in \mathcal{W} \setminus \mathcal{W}'$ *and* $L(\mathcal{G}, M \cup \{u, v, (u, v)\}) < L(\mathcal{G}, M)$ **do**

6      $\lfloor$   $\mathcal{W}' \leftarrow \mathcal{W}' \cup (u, v)$; $M \leftarrow M \cup \{u, v, (u, v)\}$      ▷ Select the candidate edges until no additional edge reduces MDL

7   $\tilde{\mathbf{A}} \leftarrow$ normalized transition matrix of rule graph

8   $\mathcal{F}_{K_1} \leftarrow \mathcal{F}.\text{topk}\left(\cos(q, \mathcal{F}), K_1\right)$      ▷ Compute the cosine similarity between $q$ and $\mathcal{F}$ to get the Top-$K_1$ events

9   Map $\mathcal{F}_{K_1}$ into rule graph and compute the seeded distribution $\boldsymbol{\gamma}$

10   $\boldsymbol{\pi}^{(0)} \leftarrow \boldsymbol{\gamma}$; $k \leftarrow 0$

11   **while** $\|\alpha\boldsymbol{\gamma} + (1 - \alpha)\,\boldsymbol{\pi}^{(k)}\tilde{\mathbf{A}} - \boldsymbol{\pi}^{(k)}\|_1 > \epsilon$ **do**

12      $\lceil$   $\boldsymbol{\pi}^{(k+1)} \leftarrow \alpha\boldsymbol{\gamma} + (1 - \alpha)\,\boldsymbol{\pi}^{(k)}\,\tilde{\mathbf{A}}$

13      $\lfloor$   $k \leftarrow k + 1$      ▷ Conduct the PPR computation until the error is smaller than $\epsilon$

14   $\mathcal{U}_{\text{top}} \leftarrow \mathcal{U}.\text{topk}(\boldsymbol{\pi}, K_2)$      ▷ Obtain the Top-$K_2$ rule nodes

15   $\mathcal{F}_{\text{cand}} \leftarrow \bigcup_{u \in \mathcal{U}_{\text{top}}} \text{supp}(u)$      ▷ Collect the events mapped into Top-$K_2$ rule nodes

16   **return** $\mathcal{F}_{\text{cand}}.\text{topk}\left(\cos(q, \mathcal{F}_{\text{cand}}), K_1\right)$      ▷ Return the Top-$K_1$ events by the cosine similarity measurement

---

sparser, time-aligned rule graph and then searches for answers along rule-node neighborhoods that match the query's temporal constraints. As illustrated step by step in Fig. 2, we first assign each entity structural type labels mined from frequent relation patterns (step 1). Based on these labels, we group each event $(s, r, o, t)$ into a candidate rule node $\langle c_s, r, c_o \rangle$, so that a rule node represents an event category rather than a single event (step 2). We then build candidate edges between rule nodes that share components and repeatedly occur close in time, indicating that one type of event is typically followed or preceded by another. Next, we apply a Minimum Description Length criterion to keep only the rule nodes and edges that best explain these event trends, yielding a compact rule graph (step 3). Finally, at query time, we seed this rule graph using the query's semantic similarity and run personalized PageRank to retrieve events that are both semantically relevant and temporally consistent with the question, producing a concise set of contexts for generation. The overall procedure is summarized in Alg. 1.

## 3.2   RULE GRAPH BUILDING

During the retrieval process, working directly on raw temporal events $f = (s, r, o, t)$ can explode the search space and hide regular temporal patterns. Therefore, we first summarize events into rule nodes and then connect them with time-sensitive edges learned from data. This rule graph enables us to preserve the temporal "what-follows-what" signal and support efficient propagation during query execution.

**Entity Labeling.** Since entity categories are often unavailable, and an entity's category is largely determined by the relations it participates in (Zhang et al., 2024a), we first abstract the interactions between entities and generate the labels for entities. The goal is to replace various entities with reusable patterns, reducing the density of the graph and minimizing the search space during retrieval. Concretely, we collect the set of relations for every entity it participates in and apply the Apriori algorithm (Agrawal & Srikant, 1994) to mine frequent relation subsets. Then the primary combinations of relations chosen from these relation subsets are utilized to map the entity $s \in \mathcal{E}$ to a label set $\mathcal{C}_s$. To simplify the presentation, we leave the detailed algorithm in Appendix A. As shown in Step 1 of Fig. 2, the relations associated with each entity are collected to mine frequent relation subsets.[2] For instance, the entity *Oman* is identified as being strongly connected with the relation subsets *(intend_cooperate, sign_agreement, provide_aid)* and *(negotiate, support, reject_proposal)*. These subsets are then assigned specific identifiers, such as $Country_1$ and $Country_2$, which serve as the labels of *Oman*.

---

[2] To control label complexity, we restrict the subset length to three relations by default.

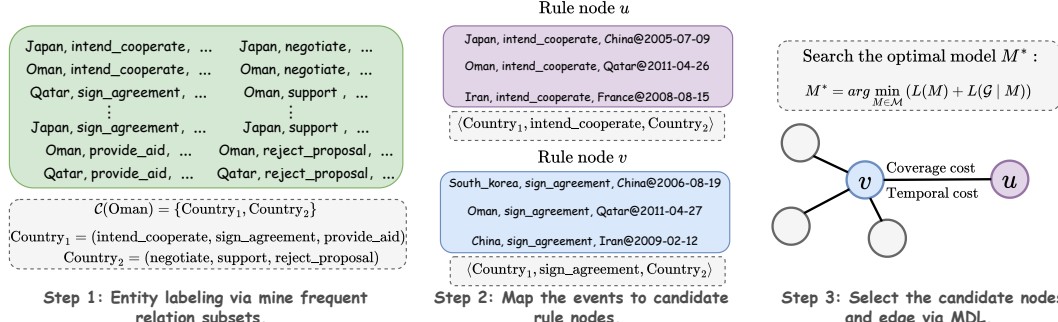

Figure 2: Running example of building the rule graph based on *MultiTQ*.

**Generation of Candidates.** *(i) Candidate nodes.* After assigning the types for entities, we further group each temporal event $(s, r, o, t)$ into one or, when applicable, multiple rule nodes:
$$\phi(s, r, o) = \{\langle c_s, r, c_o \rangle \mid c_s \in \mathcal{C}(s), c_o \in \mathcal{C}(o)\}, \tag{1}$$
where each $u = \langle c_s, r, c_o \rangle$ is regarded as a candidate rule node to summarize the pattern of a series of events, and and the collection of rule nodes with different $c_s$ and $c_o$ captures the multiple dynamic roles that entities can assume over time. All rule nodes define the candidate rule set $\mathcal{U} = \{u | \exists (s, r, o, t) \in \mathcal{F} \text{ with } u \in \phi(s, r, o)\}$ with a support set $\mathrm{supp}(u) = \{(s, r, o, t) | (s, r, o, t) \in \mathcal{F} \text{ and } u \in \phi(s, r, o)\}$. As shown in the example presented by Fig. 2, a series of events containing relation *intend_cooperate* are mapped into rule node $u$ when the subject and object belong to *Country*$_1$ and *Country*$_2$ respectively.

*(ii) Candidate edges.* Questions on temporal knowledge graphs typically target time-stamped relations among entities (Chen et al., 2023; Saxena et al., 2021), e.g., "After establishing diplomatic cooperation with Qatar, what did Oman immediately do?" This calls for modeling temporal patterns across rule nodes and linking strongly related nodes. To achieve this, we exploit shared components within rule nodes and model how event categories tend to follow one another. Given two rules $u = \langle a_s, r, a_o \rangle$ and $v = \langle b_s, r', b_o \rangle$, we define the Hamming distance between $u$ and $v$ as:
$$d_H(u, v) = \mathbf{1}\left[a_s \neq b_s\right] + \mathbf{1}\left[r \neq r'\right] + \mathbf{1}\left[a_o \neq b_o\right]. \tag{2}$$
And the candidate edge set can be defined as $\mathcal{W} = \{\{u, v\} : u \neq v, d_H(u, v) \leq 1\}$. Establishing such an edge captures the intuition that rules sharing multiple fields are likely to encode strongly related event patterns. This construction enriches the connectivity of the rule graph and facilitates effective propagation from a given anchor event to other semantically close patterns. For example, considering the anchor event $(Oman, intend\_cooperate, Qatar, 2011\text{-}04\text{-}26) \in \mathrm{supp}(u)$ in Fig. 2, the built edge $(u, v)$ guides the search from $u$ to the neighboring rule node $v$, which may contain a critical clue for answering the query, such a $(Oman, sign\_agreement, Qatar, 2011\text{-}04\text{-}27)$. Additionally, we also measure the temporal proximity between two connected rule nodes. Formally, given the events $f = (s, r, o, t) \in \mathrm{supp}(u)$ and $f' = (s', r', o', t') \in \mathrm{supp}(v)$, we collect the time span between $u$ and $v$ as $\mathcal{T}_{uv} = \left\{|t' - t| \big| f \in \mathrm{supp}(u), f' \in \mathrm{supp}(v), d_H(\tilde{f}, \tilde{f}') \leq 2\right\}$, where we reuse the Hamming distance from Equ.2 to measure the overlap between two events[3]. This set of temporal spans captures the temporal differences between highly similar events across $u$ and $v$, which is subsequently incorporated into the final edge selection process.

**Selection by MDL.** The two-part MDL principle (Rissanen, 1978) states that, given a family of models $\mathcal{M}$, the best model $M^* \in \mathcal{M}$ for graph $\mathcal{G}$ is the one minimizing:
$$M^* = \arg \min_{M \in \mathcal{M}} L(\mathcal{G}, M) = \arg \min_{M \in \mathcal{M}} \left(L(M) + L(\mathcal{G} \mid M)\right), \tag{3}$$
where $L(M)$ is the cost of describing the model itself and $L(\mathcal{G} \mid M)$ is the cost of describing the graph under that model. In this work, we adopt the MDL principle to identify the optimal rule model $M$ for the graph $\mathcal{G}$. To simplify the presentation, we follow the standard formulation utilized in Zhang et al. (2024a) to calculate $L(M)$, with full derivations and details provided in Appendix B.

We decompose the total cost of describing $\mathcal{G}$ as
$$L(\mathcal{G} \mid M) = \underbrace{L_{\mathrm{cov}}(\mathcal{G} \mid M)}_{\text{Coverage cost}} + \underbrace{L_{\mathrm{time}}(\mathcal{G} \mid M)}_{\text{Temporal cost}}. \tag{4}$$

---

[3]For events, we relax to $d_H \leq 2$ as timestamps are always different.

(i) $L_{\text{cov}}(\mathcal{G} \mid M)$ encodes how many strongly related events are explained by the selected edges. We define this cost function as:

$$L_{\text{cov}}(\mathcal{G} \mid M) = \sum_{\{u,v\} \in M} \log \left( \frac{|\text{supp}(u)| \cdot |\text{supp}(v)|}{|\mathcal{T}_{uv}|} \right). \quad (5)$$

Thus, adding an edge that explains more connected events will bring fewer costs, rewarding high-support, high-conversion links.

(ii) Compared with the coverage cost, $L_{\text{time}}(\mathcal{G} \mid M)$ quantifies the degree to which the observed temporal spans align with the temporal behavior of each rule edge. For an edge $(u, v)$ with $|\mathcal{T}_{uv}|$ spans, we assume they follow an exponential distribution with maximum-likelihood rate $\lambda_{uv} = \frac{|\mathcal{T}_{uv}|}{\sum_{d \in \mathcal{T}_{uv}} d}$. The negative log-likelihood code length is then

$$L_{\text{time}}(u, v) = - \sum_{d \in \mathcal{T}_{uv}} \log\left(\lambda_{uv} e^{-\lambda_{uv} d}\right) = |\mathcal{T}_{uv}| + |\mathcal{T}_{uv}| \log\left( \frac{1}{|\mathcal{T}_{uv}|} \sum_{d \in \mathcal{T}_{uv}} d \right). \quad (6)$$

We sum across edges as the final temporal cost $L_{\text{time}}(\mathcal{G} \mid M) = \sum_{(u,v) \in M} L_{\text{time}}(u, v)$. An edge $(u, v)$ is accepted if including it decreases the total description length:

$$\Delta L = \Delta L_{\text{cov}} + \Delta L_{\text{time}} < 0. \quad (7)$$

We employ a greedy strategy for edge insertion: candidate edges are examined in a fixed order, temporarily added, and retained only if their inclusion decreases the overall description length. This process repeats iteratively until no additional edge reduces the MDL.

### 3.3 Retrieval with Seeded personalized PageRank

During the retrieval process, we first compute the cosine similarity[4] between the query $q$ and the events and obtain the top-$K_1$ anchor events as $\mathcal{F}_{K_1} = \{f_1, \ldots, f_{K_1}\}$. By mapping the anchor events to the rule nodes, we can expand along the events that empirically occur close in time, surfacing neighborhoods where the right answer is likely to lie at the right time. However, distributing anchor events across multiple rule nodes and weighting them equally in propagation will cause two issues. First, it can steer the search toward rare, low-support rules, making propagation unreliable (Galárraga et al., 2013). Second, it can place unnecessary weight on rule nodes that are only weakly related to the query semantics, pulling in unrelated events.

To address these two limitations, we determine the personalization distribution with two complementary signals: *(i) Corpus coverage.* Rules that explain more events in the corpus are stronger hubs and make propagation more reliable. This favors rule nodes with larger support, which are more likely to summarize the common pattern of events. *(ii) Ranking importance.* Higher-ranked anchors receive larger weights, and rule nodes reached by top anchors are assigned extra probability mass to reflect semantic relevance to the query. As a result, we construct a personalization vector $\boldsymbol{\gamma}$ over the seeded rule nodes $\mathcal{U}_{\text{seed}}$ with $\|\boldsymbol{\gamma}\|_1 = 1$, which serves as the starting distribution in PPR. The computation of $\boldsymbol{\gamma}$ is detailed in Appendix C. We also conduct an ablation study (see Sec. 4.5) that replaces $\boldsymbol{\gamma}$ with the uniform distribution over $\mathcal{U}_{\text{seed}}$, which highlights the benefit of our design.

Let $\tilde{\mathbf{A}}$ denote the transition matrix of our rule graph formed by building edges and normalizing edge weights. With personalization vector $\boldsymbol{\gamma}$ over $\mathcal{U}_{\text{seed}}$, we then diffuse the personalization vector with personalized PageRank:

$$\boldsymbol{\pi} = \alpha \boldsymbol{\gamma} + (1 - \alpha) \boldsymbol{\pi} \tilde{\mathbf{A}}, \quad (8)$$

where $\alpha$ is the decay coefficient (set as 0.2 by default). We run standard power iteration until convergence within a fixed tolerance error $\epsilon$ and take the top-$K_2$ rules by $\boldsymbol{\pi}$. We collect the events associated with these rules and re-rank them by cosine similarity to the query, returning the top-$K_1$ events as the final retrieval set. By combining the retrieval results with our prompt templates, the LLM input preserves semantic precision while leveraging time-aware exploration to prioritize temporally proximate events. We leave our used prompt templates and a running example in Appendix E.

## 4 Experiments

### 4.1 Datasets

We conduct our experiments on three real-world temporal knowledge graph datasets widely used for temporal question answering: *CronQuestion* (Saxena et al., 2021), *Forecast* (Ding et al., 2023),

---

[4]We pre-compute the embedding of events and queries with the embedding model NV-Embed (Lee et al.)

and *MultiTQ* (Chen et al., 2023). CronQuestion is constructed from all temporally annotated facts in Wikidata (Lacroix et al., 2020), with hundreds of templates employed for question generation. Similarly, Forecast and MultiTQ derive temporal events from the ICEWS21 and ICEWS14 data streams[5], where questions are generated by filling templates with entity aliases. Each dataset contains over 300K events and a large set of questions requiring complex topological and temporal reasoning, posing non-trivial challenges for existing GraphRAG methods. To further evaluate the performance on large-scale KGs, we reconstructed a larger-scale evaluation corpus from ICEWS 2005–2015, 2018, and 2021, covering over 1.2 millions events, named as *STAR-QA*, where the details can be found in the Appendix I. To limit experimental cost, we randomly sample 1,000 questions from the test set of each dataset and report the average performance over five runs. The detailed statistics of three datasets are introduced in Table 1.

## 4.2 BASELINE METHODS

We compare STAR-RAG with a set of representative GraphRAG or temporal KG baselines. For static knowledge graphs, we include TOG (Sun et al., 2024), Medical-GraphRAG (Wu et al., 2024b), G-Retriever (He et al., 2024), DALK (Li et al., 2024), and HippoRAG (Gutiérrez et al., 2024). For temporal settings, we include TS-Retriever (Wu et al., 2024a), T-GRAG (Li et al., 2025), TimeR4 Qian et al. (2024), and DyG-RAG Sun et al. (2025) as state-of-the-art temporal GraphRAG methods. A detailed introduction is provided in Appendix F.1.

Table 1: Statistics of the datasets. $|\mathcal{F}|$, $|\mathcal{E}|$, $|\mathcal{R}|$, $|\mathcal{T}|$ denote the numbers of events, entities, relations, and valid timestamps respectively.

| **Datasets** | $|\mathcal{F}|$ | $|\mathcal{E}|$ | $|\mathcal{R}|$ | $|\mathcal{T}|$ |
|---|---|---|---|---|
| *CronQuestion* | 328,635 | 125,726 | 203 | 1,643 |
| *Forecast* | 335,303 | 20,575 | 253 | 243 |
| *MultiTQ* | 461,329 | 10,488 | 251 | 4,017 |
| *STAR-QA* | 1,265,132 | 34,945 | 265 | 4,261 |

## 4.3 IMPLEMENTATION DETAILS

By default, we use NV-Embed (Lee et al.) to encode all events into vector representations for computing semantic similarity across all methods. We employ `Llama-3.3-70b-instruct-awq` (Hansen & Meta, 2024) as the LLM backbone by default for all methods. For STAR-RAG, we set the hyperparameters to $K_1 = 10$ and $K_2 = 20$, corresponding to the retrieved events and the number of ranked rule nodes respectively in the PPR-based retrieval pipeline. Moreover, we set the tolerance error $\epsilon$ as $10^{-5}$ for the fast convergence of PPR. In terms of compute requirements, our experiments are conducted on a Linux server equipped with an Intel(R) Xeon(R) Silver 4314 CPU @ 2.40 GHz, 500 GB RAM, and 4× NVIDIA RTX A30 GPUs (24 GB each).

## 4.4 MAIN RESULTS

Following prior work (Gutiérrez et al., 2024; Wu et al., 2024a), we evaluate using the Hit@$k$ metric, where Hit@$k$ denotes the proportion of correct answers appearing among the top-$k$ answer results. For example, Hit@1 reflects strict top-rank accuracy, while Hit@5 measures whether the correct answer is included within the top five answers. For fair comparison, we tune the hyperparameters of each baseline and report their best achievable performance.

**Superior Performance over Baselines.** Table 2 shows that our method consistently outperforms all baselines across the three datasets. Static-graph approaches such as TOG and DALK perform notably worse than temporal methods, as they rely solely on semantic matching between queries and events while neglecting temporal patterns, which leads to low retrieval effectiveness. Remarkably, STAR-RAG achieves at least a 5% improvement in Hit@1 over the strongest baseline TS-Retriever, and the gains are similarly evident for Hit@5 and Hit@10. These improvements stem from STAR-RAG's ability to construct a time-aligned search space over the knowledge graph. By prioritizing events that occur close in time to the anchor events, STAR-RAG surfaces key clues within the retrieval candidates, alleviates the filtering burden on the LLM, and ultimately improves retrieval quality.

**Remarkable capability for reasoning over complex questions.** To further examine the rationale behind STAR-RAG's effectiveness on temporal questions, we separate the question types of *MultiTQ*

---

[5]https://dataverse.harvard.edu/dataverse/icews

Table 2: The accuracy (%) of question answering on three datasets. The best results are bold, and "OOM" stands for out of memory on 4 GPUs with 24GB memory.

| Method | CronQuestion | | | Forecast | | | MultiTQ | | | STAR-QA | | |
|---|---|---|---|---|---|---|---|---|---|---|---|---|
| | Hit@1 | Hit@5 | Hit@10 | Hit@1 | Hit@5 | Hit@10 | Hit@1 | Hit@5 | Hit@10 | Hit@1 | Hit@5 | Hit@10 |
| TOG (Sun et al., 2024) | 69.5 | 76.4 | 77.2 | 29.3 | 31.5 | 33.4 | 22.6 | 32.4 | 34.5 | 21.6 | 29.1 | 30.4 |
| MedicalGraphRAG (Wu et al., 2024b) | 50.4 | 67.2 | 73.3 | 30.4 | 41.2 | 47.8 | 21.1 | 31.4 | 36.3 | 30.2 | 34.3 | 35.7 |
| DALK (Li et al., 2024) | 58.1 | 71.0 | 73.6 | 28.6 | 32.7 | 36.4 | 17.1 | 28.4 | 39.7 | 21.7 | 28.0 | 28.9 |
| G-Retriever (He et al., 2024) | 19.7 | 28.4 | 34.7 | 12.1 | 23.8 | 25.8 | 9.5 | 18.6 | 20.9 | 9.2 | 12.3 | 16.6 |
| HippoRAG (Gutiérrez et al., 2024) | 54.2 | 69.7 | 75.5 | 29.1 | 36.2 | 40.8 | 18.2 | 29.1 | 39.9 | 18.3 | 30.1 | 36.8 |
| TS-Retriever (Wu et al., 2024a) | 68.5 | 74.1 | 75.6 | 32.1 | 44.4 | 49.7 | 25.5 | 36.2 | 40.3 | 31.2 | 38.4 | 42.1 |
| T-GRAG (Li et al., 2025) | 67.3 | 72.9 | 74.0 | 31.2 | 42.3 | 47.7 | 25.2 | 35.1 | 41.5 | 32.0 | 38.1 | 41.6 |
| TimeR4 Qian et al. (2024) | 65.7 | 73.5 | 77.8 | 30.5 | 41.3 | 45.9 | 23.4 | 31.3 | 37.4 | OOM | OOM | OOM |
| DyG-RAG Sun et al. (2025) | 64.3 | 70.9 | 72.4 | 29.5 | 39.3 | 44.5 | 23.3 | 32.1 | 38.8 | OOM | OOM | OOM |
| STAR-RAG | **76.9** | **85.4** | **87.0** | **39.8** | **51.4** | **55.7** | **30.5** | **41.5** | **47.7** | **36.8** | **45.4** | **51.9** |

into the single-event and multiple-event categories, where the results are shown in Table 3. Here, the single-event questions require only one fact to obtain the answer by identifying the entity, relation, or timestamp in a single interaction, such as "Who asked for the government of Sudan in 2010?". In this setting, static GraphRAG systems can solve the task through straightforward semantic matching, reflected in their competitive performance. For instance, MedicalGraphRAG achieves the best accuracy at 26.7% Hit@1. By contrast, multiple-event questions demand compositional reasoning over event chains, where the answer depends on an anchor interaction and subsequent temporal constraints, such as "After Mallam Isa Yuguda, with whom did USAID first formally sign an agreement?". We observe that static methods degrade sharply in this regime because they retrieve large, temporally mixed candidate sets, leaving the LLM with a heavy filtering burden.

Temporal baselines partially mitigate this issue, yet their improvements remain limited. Fortunately, STAR-RAG achieves a clear jump to 44.4% Hit@1 on multiple-event questions, nearly doubling the best competing system at 23.8% of T-GRAG. This gap indicates that building a time-aligned rule graph and propagating around identified anchors narrows the search to time-consistent evidence, reduces semantic confusion, and yields more faithful answers under temporal constraints. In summary, STAR-RAG preserves single-event performance while delivering large gains on multi-event reasoning, suggesting improved temporal reasoning capability without sacrificing basic semantic matching.

Table 3: Accuracy (%) across different question types. Results are reported on the *MultiTQ* dataset using the Hit@1 metric.

| Method | Single-event | Multiple-event |
|---|---|---|
| TOG | 25.3 | 12.6 |
| MedicalGraphRAG | **26.7** | 4.5 |
| DALK | 21.3 | 4.9 |
| G-Retriever | 10.7 | 4.3 |
| HippoRAG | 22.6 | 5.4 |
| TS-Retriever | 26.8 | 21.7 |
| T-GRAG | 25.7 | 23.8 |
| TimeR4 | 24.3 | 19.2 |
| DyG-RAG | 25.0 | 18.4 |
| STAR-RAG | 25.8 | **44.4** |

**Lower token consumption and comparable reasoning time.** To evaluate the efficiency of STAR-RAG, we measure the average token consumption and reasoning time of each method on the *MultiTQ* dataset, as shown in Fig. 3. The results on the other two datasets can be found in Appendix D. Static methods such as TOG and MedicalGraphRAG consume little reasoning time, since they rely purely on vector similarity for retrieval. While this yields fast but coarse results, it comes at the cost of higher token usage and reduced accuracy. In contrast, temporal methods like TS-Retriever and T-GRAG explicitly enforce temporal constraints and search for time-aligned events, which improves accuracy but requires substantially more computation time. STAR-RAG achieves a more favorable balance between efficiency and effectiveness: it reduces token usage by up to (20453-601)/20453 = 97.0% compared with MedicalGraphRAG, with only about 10 seconds of additional reasoning time. This demonstrates that STAR-RAG can significantly lower the LLM's token burden while maintaining practical inference speed.

## 4.5 ABLATION STUDIES

In this section, we perform ablation studies on the variants of STAR-RAG to assess the contribution of each module, as summarized in Table 4.

**STAR-RAG is robust to the LLM backbones.** We replace the default LLM generator with `Llama-3-8B-Instruct` (AI@Meta, 2024) and `GPT-4o-mini` (Achiam et al., 2023) to evaluate the impact of backbone choices on prediction accuracy and reasoning time. The results show that

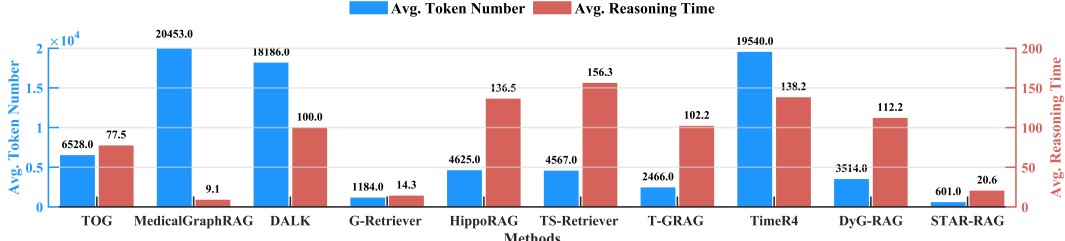

Figure 3: Comparison of token consumption and reasoning time based on *MultiTQ*.

Table 4: The average accuracy (%, denoted as Acc.) and reasoning time (s, denoted as Tim.) for each variant of STAR-RAG.

| Method | CronQuestion | | Forecast | | MultiTQ | |
|---|---|---|---|---|---|---|
| | Acc. | Tim. | Acc. | Tim. | Acc. | Tim. |
| STAR-RAG+Llama-3-8B-Instruct | 73.4 | 27.5 | 35.4 | 20.0 | 26.5 | 18.7 |
| STAR-RAG+GPT-4o-mini | 74.3 | 28.0 | 36.9 | 20.2 | 28.6 | 18.8 |
| STAR-RAG-no-rule | 55.3 | 15.6 | 29.8 | 10.2 | 18.3 | 10.5 |
| STAR-RAG-uniform | 70.0 | 29.4 | 31.1 | 20.6 | 22.7 | 19.2 |
| STAR-RAG | 76.9 | 30.4 | 39.8 | 22.2 | 30.5 | 20.6 |

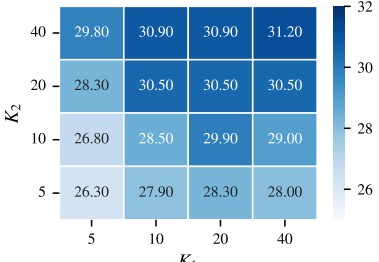

Figure 4: Performance of setting different $K_1$ and $K_2$ based on *MultiTQ*.

changing the LLM backbone leads to only modest performance degradation, with a maximum drop of $4.4\%$ even when using the lightweight `Llama-3-8B-Instruct`. These consistent margins indicate that the retrieval pipeline provides most of the performance gains, ensuring that STAR-RAG remains effective across different LLM generators.

**Rule graph substantially improves retrieval quality.** We ablate the search mechanism of the rule graph and replace it with semantic search alone (denoted as STAR-RAG-no-rule in Table 4). The results show that removing the rule graph leads to significant accuracy losses compared with the complete STAR-RAG ($21.6\%$, $10.0\%$, and $12.2\%$ degradation on *CronQuestion*, *Forecast* and *MultiTQ* for Hit@1), even though it roughly halves the inference time. This highlights that time-aligned propagation over the rule graph is crucial for selecting high-quality evidence.

**Corpus coverage and ranking importance prioritize the events most relevant to questions.** Finally, we replace the personalization vector $\gamma$ with a uniform distribution over the rule nodes (denoted as STAR-RAG-uniform), which results in up to an $8.7\%$ drop in Hit@1 on the *Forecast* dataset. This shows that weighting seeds by corpus coverage and anchor rank helps propagation focus on rules that explain more events and align with the query, which yields more reliable evidence.

### 4.6 SENSITIVITY TO THE NUMBER OF RETRIEVED RULE NODES AND EVENTS

We change the values of $K_1, K_2 \in \{5, 10, 20, 40\}$ and report Hit@1 for each setting (Fig. 4). We observe that accuracy improves as both parameters increase, with a stronger effect from $K_2$: raising $K_2$ from 10 to 20 yields a clear gain, while further increasing to 40 provides only marginal benefit. Raising $K_1$ improves accuracy at first but then quickly plateaus, suggesting that broader rule coverage matters more than simply adding more events, which can introduce noise. In terms of cost, a larger $K_2$ lengthens the retrieval process due to the semantic matching over more rule nodes, while a larger $K_1$ increases the LLM's token budget and burdens generation. After balancing accuracy, latency, and token usage, we adopt the default $K_1$=10 and $K_2$=20, which provides a balance between accuracy and efficiency.

### 5 CONCLUSION

Existing RAG systems built on static KGs or texts often fail to achieve time-aligned retrieval when faced with temporal reasoning tasks. To address this limitation, we present STAR-RAG, a temporal GraphRAG framework that summarizes the temporal knowledge graph into a concise rule graph and leverages seeded personalized PageRank to propagate evidence along temporally consistent paths. STAR-RAG provides efficient and accurate answers without requiring additional training and demonstrates robustness across a range of LLM backbones. Results on three temporal KG datasets confirm that STAR-RAG surpasses both static and temporal GraphRAG baselines, offering superior accuracy with higher token efficiency.

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

## A  DETAILED LABELING ALGORITHM

---

**Algorithm 2:** ENTITYLABELING

---
**Input:** Event set $\mathcal{F} = \{(s, r, o, t)\}$; graph $\mathcal{G}$
**Output:** Labels $\mathcal{C}(s)$ for each $s \in \mathcal{E}$
1 Build $\mathrm{RelSet}[s] := \{r \mid (s, r, *, *) \text{ or } (*, r, s, *) \in \mathcal{F}\}$ for all $s \in \mathcal{E}$  $\quad\triangleright$ Construct the relation set for entities
2 $\mathcal{F} \leftarrow \mathrm{APRIORI}([\,\mathrm{RelSet}[e]\,]_{e \in \mathcal{E}})$  $\quad\triangleright$ Apply Apriori algorithm mine frequent relation subsets
3 Sort $\mathcal{F}$ and assign the TypeID for each combination of relations
4 **foreach** $s \in \mathcal{E}$ **do**
5 $\quad L \leftarrow [\,\mathrm{TypeID}[s] \mid p \in \mathcal{F},\ p \subseteq \mathrm{RelSet}[s]\,]$
6 $\quad$ **if** $L$ *is empty* **then**
7 $\quad\quad p^\star \leftarrow \mathrm{RelSet}[s]$
8 $\quad\quad L \leftarrow [\,\mathrm{TypeID}[p^\star]\,]$  $\quad\triangleright$ We treat the relation set of $s$ as $L$ if $L$ is empty
9 $\quad \mathcal{C}(s) \leftarrow$ the first $K_{\mathrm{type}}$ combinations in $L$  $\quad\triangleright$ We only keep the Top-$K_{\mathrm{type}}$ combinations as the labels of $s$
10 **return** $\mathcal{C}(s)$ for each $s \in \mathcal{E}$

---

## B  DETAILS OF COMPUTING MODEL COST OF THE RULE GRAPH

**Notations.**  Let $\mathcal{G} = (\mathcal{E}, \mathcal{R}, \mathcal{T}, \mathcal{F})$ be the TKG. Let $\mathcal{A}$ be the finite set of category labels and $\mathcal{C} : \mathcal{E} \to \mathcal{A}$ map each entity to a label; write $A := |\mathcal{A}|$. Our rule graph is $M = (\mathcal{U}, \mathcal{E}_{\mathrm{rule}})$ with nodes $u = \langle a_s, r, a_o \rangle \in \mathcal{A} \times \mathcal{R} \times \mathcal{A}$ and directed *chain* edges $(u \to v)$. Candidate edges are restricted to Hamming-1 neighbors:
$$\mathcal{W} = \big\{\{u, v\} : u \neq v,\ d_H(u, v) \leq 1\big\}, \quad d_H(u, v) = \mathbf{1}[a_s \neq b_s] + \mathbf{1}[r \neq r'] + \mathbf{1}[a_o \neq b_o].$$

**What $L(M)$ measures.**  $L(M)$ counts the bits to (i) choose nodes from all atomic-rule options, (ii) choose *directed* chain edges from admissible pairs, and (iii) encode the selected nodes and edges using optimal prefix codes from empirical frequencies.

We adopt the two-part MDL form $L(M) + L(\mathcal{G} \mid M)$ and detail $L(M)$:

$$L(M) = \underbrace{\log_2\!\big(A^2 |\mathcal{R}|\big)}_{\text{candidate atomic rules}} + \underbrace{\log_2\!\big(2\,|\mathcal{W}|\big)}_{\text{candidate \emph{directed} chain edges}} + \sum_{u \in \mathcal{U}} L(u) + \sum_{(u \to v) \in \mathcal{E}_{\mathrm{rule}}} L(u \to v).$$

**Node code length.**  From $\mathcal{F}$, estimate empirical probabilities $p_s(a)$ and $p_o(a)$ for subject/object categories $a \in \mathcal{A}$, and $p_r(r)$ for relations $r \in \mathcal{R}$. Then
$$L(u) = -\log_2 p_s(a_s)\ -\ \log_2 p_r(r)\ -\ \log_2 p_o(a_o), \quad u = \langle a_s, r, a_o \rangle.$$

**Edge code length.**  Let $d^{\mathrm{out}}(u)$ and $d^{\mathrm{in}}(u)$ be the out-/in-degrees of $u$ in $\mathcal{E}_{\mathrm{rule}}$ and define the endpoint distribution
$$p_V(u) = \frac{d^{\mathrm{out}}(u) + d^{\mathrm{in}}(u)}{2\,|\mathcal{E}_{\mathrm{rule}}|}.$$
A directed chain edge $(u \to v)$ is encoded as
$$L(u \to v) = -\log_2 p_V(u)\ -\ \log_2 p_V(v).$$

## C  DETAILED ALGORITHM OF COMPUTING PERSONALIZATION VECTOR

**Seeded personalization vector $\gamma$.**  Let $\mathcal{F}_{K_1} = \{f_1, \ldots, f_{K_1}\}$ be the Top-$K_1$ anchors for query $q$ (ordered by cosine similarity). We map $\mathcal{F}_{K_1}$ to rule nodes and form the seeded rule set
$$\mathcal{U}_{\mathrm{seed}} = \big\{u \in \mathcal{U} : \mathrm{supp}(u) \cap \mathcal{F}_{K_1} \neq \emptyset\big\}.$$

*Corpus coverage (counts).* For each $u \in \mathcal{U}_{\mathrm{seed}}$, define the raw coverage $c_u = |\mathrm{supp}(u)|$ and its normalization
$$\tilde{c}_u = \frac{c_u}{\sum_{v \in \mathcal{U}_{\mathrm{seed}}} c_v}.$$

*Ranking importance (geometrically discounted hits).* Let $\beta \in (0, 1)$ and write $f_j$ for the $j$-th anchor (rank $j$ is 1-based). Define

$$p_u = \sum_{j:\, f_j \in \mathrm{supp}(u)} \beta^{j-1}, \qquad \tilde{p}_u = \frac{p_u}{\sum_{v \in \mathcal{U}_{\mathrm{seed}}} p_v}.$$

*Blending and smoothing.* With mixture weight $\theta \in [0, 1]$ and Dirichlet smoothing $\tau > 0$,

$$s_u = (1 - \theta)\, \tilde{c}_u + \theta\, \tilde{p}_u, \qquad \gamma_u = \frac{s_u + \tau}{\sum_{v \in \mathcal{U}_{\mathrm{seed}}} (s_v + \tau)}, \quad u \in \mathcal{U}_{\mathrm{seed}}.$$

In our implementation we set $\theta = 0.6$, $\beta = 0.7$, and use $\tau = 1/|\mathcal{U}_{\mathrm{seed}}|$ by default. Finally, we diffuse $\boldsymbol{\gamma}$ on the rule graph by personalized PageRank:

$$\boldsymbol{\pi} = \alpha\, \boldsymbol{\gamma} + (1 - \alpha)\, \boldsymbol{\pi}\, \tilde{\mathbf{A}},$$

where $\alpha = 0.2$ is the restart probability used in our experiments.

# D  ADDITIONAL EXPERIMENTAL RESULTS

Here, we present the comparison of token consumption and reasoning time based on *CronQuestion* and *Forecast* datasets in Fig. 5 and 6.

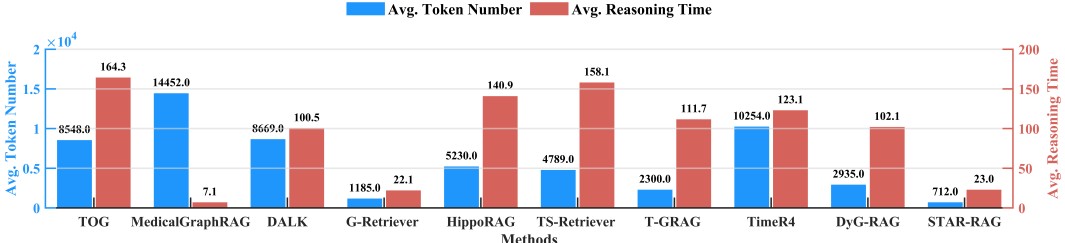

Figure 5: Comparison of token consumption and reasoning time on *CronQuestion*.

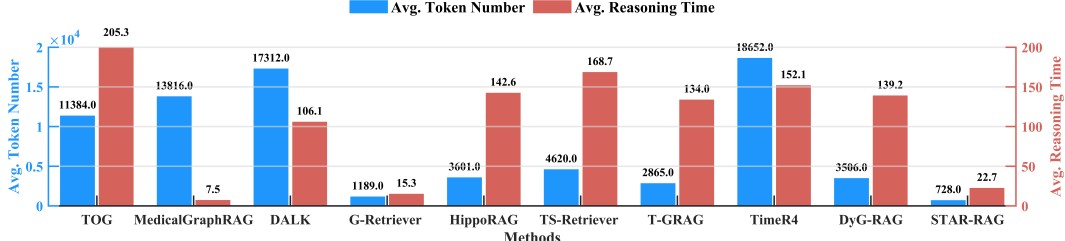

Figure 6: Comparison of token consumption and reasoning time on *Forecast*.

Then we report the rule-graph construction time and memory usage, as well as the per-query reasoning time for each dataset, in Table 5.

Table 5: Runtime and memory measurements for STAR-RAG: rule-graph construction time (s), peak memory usage (MB), and per-query reasoning time (s) on each dataset.

| Measurements | CronQuestion | Forecast | MultiTQ | STAR-QA |
|---|---|---|---|---|
| Building Time | 254.3 | 283.9 | 268.9 | 384.6 |
| Building Memory | 1,062 | 1,232 | 1,102 | 2,754 |
| Reasoning Time | 20.6 | 23.0 | 22.7 | 35.6 |

## E  PROMPT AND EXAMPLE FOR TEMPORAL QUESTION ANSWERING

**Prompt**

**Instruction:**
As an advanced reading comprehension assistant, your task is to analyze multiple triple facts and corresponding questions with time constraints meticulously.
Your response start after "Thought: ", where you will methodically break down the reasoning process, illustrating how you arrive at conclusions.
Keep subject/object orientation. Match the same base relation. Apply temporal operator precisely.
Conclude with "Answer: " to present return 10 short answer candidates ranked best-to-worst, devoid of additional elaborations.

**One-Shot Demonstration:**
**Events:**
Event A: On 2010-08-30, European Central Bank criticized Romania. Event B: On 2011-02-14, European Central Bank criticized government of Germany.
**Question:**
Before Germany, who did the European Central Bank criticize last?
**Answer:**
Romania.

**An example to retrieve the time-aligned events**

**Question:**
Which country was the last to accuse the UN security council before the Military Personnel of Canada did?
**Retrieval results:**

1. `Military_personnel_(Canada) accuse Un_security_council @ 2006-02-10`

2. `Un_security_council accuse Government_sudan @ 2006-01-27`

3. `Un_security_council accuse Iran @ 2005-07-31`

4. `Lawmaker_(United_kingdom) accuse Government_(Sudan) @ 2006-01-26`

5. `Citizen_(Africa) accuse Government_(Sudan) @ 2006-02-04`

6. `Congress_(Philippines) accuse Military_personnel_(Philippines) @ 2006-01-30`

7. **`Eritrea accuse Un_security_council @ 2006-01-04`**

8. `Military_(Philippines) accuse Military_personnel_(Philippines) @ 2005-12-26`

9. `Military_personnel_(Thailand) accuse Sondhi_limthongkul @ 2005-11-28`

10. `Military_personnel_(Thailand) accuse Citizen_(Thailand) @ 2005-11-22`

> **Thoughts:**
> To determine which country was the last to accuse the UN Security Council before the Military Personnel of Canada did, we need to examine the dates provided for each entity accusing the UN Security Council and compare them to the date when the Military Personnel of Canada made their accusation, which is 2006-02-10.
>
> The relevant accusations against the UN Security Council are:
> - Eritrea on 2006-01-04
> - Yemane Gebremeskel on 2005-11-24
> - Foreign Affairs (Syria) on 2005-10-31
> - UN Security Council itself is not relevant as it's the one being accused.
>
> Given these dates, the last entity to accuse the UN Security Council before the Military Personnel of Canada (on 2006-02-10) was Eritrea, which did so on 2006-01-04.
> **Answer:**
> Eritrea

## F  MORE RELATED WORKS

In addition to STAR-RAG, we summarize several recent temporal RAG and temporal LLM plus KG methods. DRAGIN Su et al. (2024) proposes a dynamic retrieval-augmented generation framework that adapts its knowledge store and retrieval strategy to streaming information needs. TimeRAG Yang et al. (2025) targets time series forecasting by retrieving relevant historical segments and using an LLM to generate future trajectories. RAG4DyG Wu et al. (2024c) focuses on dynamic graphs and uses retrieval-augmented modeling to improve link prediction under temporal evolution. DynaGRAG Thakrar (2024) designs a dynamic GraphRAG pipeline that updates and queries evolving subgraphs aligned with query time. MusTQ Zhang et al. (2024b) introduces a multi step temporal question answering benchmark and model that require compositional reasoning over temporal facts.

### F.1  INTRODUCTION OF BASELINE METHODS

We compare STAR-RAG with a set of representative GraphRAG methods on static knowledge graphs, including: TOG (Sun et al., 2024), which organizes knowledge into topic-oriented subgraphs for efficient traversal; MedicalGraphRAG (Wu et al., 2024b), which augments GraphRAG with triple-linked graphs and a coarse-to-fine retrieval strategy that combines precise matching with iterative context refinement; G-Retriever (He et al., 2024), which performs graph retrieval via a Prize-Collecting Steiner Tree, supporting scalable multi-hop reasoning; DALK (Li et al., 2024), which employs a dual-level adaptive knowledge graph to balance semantic and structural reasoning; and HippoRAG (Gutiérrez et al., 2024), which incorporates a hippocampus-inspired memory mechanism to unify short- and long-term knowledge. In addition, we include four temporal GraphRAG methods: TS-Retriever (Wu et al., 2024a), which models event dynamics and temporal dependencies; T-GRAG (Li et al., 2025), which constructs time-stamped graphs with temporal query decomposition to address conflicts and redundancy; TimeR4 Qian et al. (2024), which uses a temporal knowledge graph with a trained time-aware retriever and a multi-stage retrieve–rewrite–rerank pipeline; DyG-RAG Sun et al. (2025), which constructs a dynamic event graph and performs time-aware graph traversal with an LLM.

## G  ROBUSTNESS TO SPARSE/NOISY RELATIONS AND LONG-TAIL GRAPHS

In STAR-RAG, similarity-based anchor retrieval serves as the primary robustness backbone: for each query, we first perform dense similarity search over temporal facts, and the top-$k$ anchor events

are surfaced regardless of how well they are covered by the rule graph. While some anchors may correspond to sparse or noisy relations that are weakly represented in the rule topology, the framework is explicitly designed so that the rule graph is not a single point of failure: if an anchor event is not included in any rule node or edge, STAR-RAG simply bypasses the rule graph and injects these anchors directly into the prompt template for LLM generation. In that case, STAR-RAG effectively degenerates to a standard similarity-based RAG pipeline, behaving comparably to widely used systems such as HippoRAG Gutiérrez et al. (2024), GRAG Hu et al. (2024), and REANO Fang et al. (2024). Moreover, sparse or noisy relations and tail events in long-tail graphs typically have poor connectivity to the rest of the graph, so answering queries about them often reduces to detecting a single specific fact rather than performing complex multi-hop reasoning. This regime is in fact the most favorable case for our retrieval–generation pipeline (and for static RAG baselines), as it only requires correctly retrieving the relevant anchor event and conditioning the LLM on it, instead of relying on densely connected rule structures.

## H    COMPLEXITY ANALYSIS.

Let $\mathcal{G}$ be a temporal knowledge graph with event set $\mathcal{F}$, candidate rule-node set $\mathcal{U}$, and candidate rule-edge set $\mathcal{W}$, and let $|\mathcal{F}|$, $|\mathcal{U}|$, and $|\mathcal{W}|$ denote their sizes. Let $C_{\max}$ be the maximum number of structural labels assigned to any entity. The offline rule-graph construction is executed once per dataset. Mapping each event $(s, r, o, t) \in \mathcal{F}$ into rule-node candidates $\langle c_s, r, c_o \rangle$ costs at most $\mathcal{O}(C_{\max}^2 |\mathcal{F}|)$, and generating candidate temporal edges in $\mathcal{W}$ using local time windows costs $\mathcal{O}(f_{\max}^2 |\mathcal{F}|)$, where $f_{\max}$ is the maximum number of events within such a window. Ranking and selecting rule nodes and edges under the MDL objective then adds $\mathcal{O}(|\mathcal{U}| \log |\mathcal{U}| + |\mathcal{W}| \log |\mathcal{W}|)$, so the total offline cost is
$$\mathcal{O}\big(C_{\max}^2 |\mathcal{F}| + f_{\max}^2 |\mathcal{F}| + |\mathcal{U}| \log |\mathcal{U}| + |\mathcal{W}| \log |\mathcal{W}|\big),$$
which is near-linear in $|\mathcal{F}|$ in our setting (small $C_{\max}$, $f_{\max}$ and $|\mathcal{U}|, |\mathcal{W}| \ll |\mathcal{F}|$). At query time, STAR-RAG first performs dense similarity search over all events and keeps the top-$K_1$ anchor events, which is $\mathcal{O}(|\mathcal{F}|)$ with a linear scan plus $\mathcal{O}(K_1 \log K_1)$ for selection; mapping these $K_1$ anchors to rule nodes is $\mathcal{O}(K_1 C_{\max})$. Personalized PageRank on the rule graph (with a sparse adjacency over $\mathcal{W}$) costs $\mathcal{O}(I |\mathcal{W}|)$ per query, where $I$ is the number of iterations. We then select the top-$K_2$ rule nodes and gather their supported events into a candidate set $\mathcal{F}_{\text{cand}}$ with size at most $|\mathcal{F}_{\text{cand}}| \leq K_2 \bar{s}$, where $\bar{s}$ is the average support size of a selected rule node; re-ranking these candidates against the query costs $\mathcal{O}(|\mathcal{F}_{\text{cand}}|)$. Overall, the per-query complexity can be written as
$$\mathcal{O}\big(|\mathcal{F}| + I |\mathcal{W}| + K_1 C_{\max} + K_2 \bar{s}\big),$$
and since $K_1, K_2, C_{\max}$, and $\bar{s}$ are small constants in our experiments, the dominant terms in practice are a single pass over $\mathcal{F}$ for similarity search and a sparse PPR over the much smaller rule graph, while the MDL-based construction (including ranking and selection) is amortized as a one-time offline cost.

## I    DETAILS FOR BUILDING STAR-QA

To evaluate the performance of STAR-RAG on large-scale KGs, we additionally reconstruct the evaluation data by ourselves from ICEWS 2005–2015, 2018, 2021 and augment it with 1,000 LLM-generated QA pairs that are explicitly designed to be more human-written, out-of-template, and to include adversarial temporal confounds. Following the question-generation process of FAITH Jia et al. (2024), we prompt the LLM (Claude) with event facts to synthesize new questions, but we design our own dataset-specific prompt (shown as follows) to explicitly encourage more human-written phrasings and adversarial temporal confounds such as distractor years and plausible but incorrect time periods.

**Prompt**

**Instruction:**

You are constructing a temporal QA dataset from a *series of events*. For each question, you are given:

- a unique question ID `quid` (integer);
- a list of time-stamped events, where each event has a subject entity, an object entity, a relation description, and an event date in the form `YYYY-MM-DD`;
- a correct answer entity, associated with one specific *target event* in this list (exactly one event justifies the answer).

Your task is to write *one* natural, conversational English question whose only correct answer is exactly the given answer entity, and whose meaning depends on the target event and its time. Other events in the list may be used only as contextual or adversarial distractors.

**Style and temporal requirements:**

- Rewrite names of the form `"X (Country)"` into natural phrases, e.g., "Foreign Affairs (Iran)" → "the Foreign Affairs ministry in Iran", "Police (Russia)" → "the police in Russia", "Citizen (Australia)" → "citizens of Australia".
- Rewrite dates like `"2021-01-01"` in natural English, e.g., "January 1st, 2021".
- Use conversational contractions where reasonable (e.g., "Who's" for "Who is", "gonna" for "going to"), and you may optionally add light discourse markers such as "So," or "Well," at the beginning to sound natural.
- Add adversarial temporal confounds: you may mention other events from the list or introduce extra, plausible but incorrect time expressions (e.g., another year, a broader time range, or vague phrases like "around that time") so that a model that only matches surface dates or counts mentions is misled. The true date of the target event and the correct answer must remain unchanged, and the question must still be answerable only by reasoning about the real target event and its time.

Do not invent facts that contradict any of the input events. There must be exactly one correct answer.

**Output format:**

Return your output as a single JSON object on one line, with no additional text: `{"quid": <QUID>, "question": "<YOUR_QUESTION>", "answers": ["<ANSWER_ENTITY>"], "qlabel": "Single"}`.

## J  FUTURE WORK

In this section, we also highlight a natural extension of our design as a direction for future work. At present, STAR-RAG uses a globally fixed parameter $\theta$ in the personalization vector (Appendix C), which implicitly balances temporal information (through $L_{time}$) and structural or coverage information (through $L_{cov}$) during rule-graph propagation. A more flexible variant would equip STAR-RAG with a query-adaptive controller that adjusts a query-dependent weight $\theta(q)$ according to the time sensitivity and structural complexity of the input question. This idea is consistent with recent work on time-aware retrieval-augmented generation such as TimeR[4], where the system explicitly rewrites questions to reveal temporal constraints before retrieval Qian et al. (2024), and with adaptive expert routing in retrieval-augmented Mixture-of-Experts language models such as Expert-RAG Zhou et al. (2024). Concretely, one could estimate a time-sensitivity score for each query by using simple rule-based heuristics (for example, detecting words like "before", "after", or explicit years), or by using an

optional LLM-based classifier, and then use this score to emphasize temporally aligned propagation for strongly temporal questions while giving more weight to structural and semantic coverage for more static questions. However, this design naturally brings a trade-off between computational cost and accuracy: richer controllers based on LLM scoring or expert routing can provide finer-grained adaptation but introduce additional token and latency overhead, whereas simple heuristic controllers are efficient but less expressive. In the present work we favor simplicity and efficiency by using a global $\theta$, and we leave the design and evaluation of such query-adaptive controllers, together with a systematic study of their trade-offs, as an important avenue for future research.

Another limitation of STAR-RAG is that the rule graph is constructed offline from a static temporal KG and then kept fixed, without efficient support for incremental updates when new events arrive. An important direction for future work is to develop an incremental maintenance mechanism for the rule graph, inspired by dynamic PPR-based systems such as Instant Zheng et al. (2022) and IDOL Zhu et al. (2024). In these methods, representations are updated under graph changes by locally refreshing Personalized PageRank or embeddings in the affected region instead of recomputing the entire model. Analogously, for STAR-RAG, new temporal events could update only the supports and temporal statistics of the impacted rule nodes and a small neighborhood of rule edges whose coverage or time patterns change significantly, while preserving the rest of the rule graph. This could be coupled with topology-monitorable triggers in the spirit of Instant and IDOL that decide when accumulated local changes warrant structural adjustments such as splitting or merging rule nodes or adding and pruning edges. Developing such an incremental rule-graph maintenance scheme would make STAR-RAG more suitable for frequently updated temporal KGs, while retaining its interpretable, rule-level temporal structure.

