# OpenReview forum: "Right Answer at the Right Time — Temporal Retrieval-Augmented Generation via Graph Summarization"
_ICLR.cc/2026/Conference — ICLR 2026 Conference Desk Rejected Submission_

### Official Review · Reviewer_SyuX · 2025-10-20

**Soundness:** 4
**Presentation:** 4
**Contribution:** 4
**Rating:** 8
**Confidence:** 3

**Summary:**

This paper introduces STAR-RAG, a temporal Graph RAG framework that uses a rule graph to summarize and encode event categories and temporal relations, followed up by a PPR retrieval. This approach improves accuracy against SoTA baselines, while incurring lower token lengths on three popular datasets in the domain (CronQuestion, Forecast and MultiTQ).

**Strengths:**

- This paper explores a novel and under explored domain for RAG, demonstrating retrieval while aligning with the dimensions of time and semantic context. The rule graph-based approach helps with merging MDL and temporal reasoning to provide effective solutions in the said domain. The method is mathematically grounded and the experimental results comparing with other baselines on CronQuestion, Forecast and MultiTQ advertise superiority both in terms of task success and resource consumption.
- Paper is written quite clearly and examples have been provided after each explanation to help with understanding.
- All evaluation and experiments conducted are complete and well-justified, not allowing any follow-up questions to remain un-answered. The paper covers detailed comparison against existing baselines, demonstrates capability in different complexity levels and justifies its lower reasoning token consumption and reasoning time. The ablations help with justifying the design choices made for the entire methodology.

**Weaknesses:**

Does this framework over-complicate simpler problems? While prior baselines follow a trend of higher single-event accuracy over multi-event in Table 3, STAR-RAG's behaviour is quite opposite. Do you think the candidates retrieved and passed confuses the LLM under evaluation?

**Questions:**

I was wondering if you tried other edge insertion strategies apart from a greedy approach. How do you ensure that they converge or are the most optimal way across all domains Do you think this would be dependent on the task we are using STAR-RAG for?

---

> ### Author Response · Authors · 2025-11-21
>
> # Reviewer SyuX-W1:
> Does this framework over-complicate simpler problems?
>
> ***Response:*** We sincerely thank the reviewer for the very positive assessment and high score, and we understand the concern about whether STAR-RAG might over-complicate simpler problems. The framework is designed for complex, multi-step scenarios rather than toy problems, and the additional components are necessary to achieve the robustness and accuracy.
>
>
> Our motivation is that most existing temporal RAG systems rely on substantial training or fine-tuning and therefore drift away from the original spirit of RAG, where the LLM is kept frozen and task-specific knowledge is supplied mainly through retrieval. Our goal is to restore this “retrieval-first” philosophy for temporal QA by making the graph search itself more time-focused, rather than adding more trained components on top of the LLM.
>
> Conceptually, the framework is simple. We:
> (1) group similar entities into coarse categories so that many different entities can be treated as playing the same role,
> (2) collect events that share the same pair of roles and relation into a single rule node, which represents “this kind of interaction happens frequently”, and
> (3) add a rule-graph edge when the connected rule nodes satisfy simple coverage and time conditions under the MDL criterion.
>
> This construction is done once offline and yields a sparse, time-aligned rule graph that we then use as a light routing layer on top of standard similarity search. For simpler or weakly temporal questions, the behavior of STAR-RAG is very close to a normal similarity-based RAG pipeline, so we are not adding heavy, query-time complexity just for its own sake. Instead, the extra structure is there to handle the many real temporal questions where ordering and time constraints matter, while still respecting the frozen-LLM, retrieval-centric spirit of RAG.
>
>
> # Reviewer SyuX-W2:
> While prior baselines follow a trend of higher single-event accuracy over multi-event in Table 3, STAR-RAG's behaviour is quite opposite. Do you think the candidates retrieved and passed confuses the LLM under evaluation?
>
> ***Response:*** We thank the reviewer for this insightful observation. We believe that the candidates retrieved by STAR-RAG might not confuse the LLM, especially because we keep the retrieval length short and focus on a compact set of events. The opposite trend in Table 3 can be explained by two aspects:
>
> $\bullet$ ***STAR-RAG gains more on multi-event questions.*** Existing methods are relatively strong on simpler, single-event questions but struggle on time-constrained multi-event questions, where they do not explicitly model event ordering. STAR-RAG is designed for this case. By grouping entities into roles, summarizing frequent interactions into rule nodes, and connecting them with time-sensitive edges, it retrieves more appropriate event chains, so the improvement is more visible on multi-event questions.
>
> $\bullet$ ***Dataset limitations on single-event questions.*** For single-event questions, we observed that many of them are only weakly grounded in the KG or require information that is not fully represented in the KG. In these cases, even if retrieval is improved, the LLM cannot significantly increase its accuracy. As a result, STAR-RAG behaves similarly to strong similarity-based baselines on single-event questions, while showing clearer gains where temporal and multi-event reasoning is truly required.

---

> ### Author Response · Authors · 2025-11-21
>
> # Reviewer SyuX-Q1:
> I was wondering if you tried other edge insertion strategies apart from a greedy approach. How do you ensure that they converge or are the most optimal way across all domains Do you think this would be dependent on the task we are using STAR-RAG for?
>
> ***Response:*** We thank the reviewer for this question. In our current design we use a simple MDL-greedy edge insertion strategy and we can give a clear convergence argument:
>
> $\bullet$ ***Why it converges.*** From an optimization perspective, our edge-selection procedure performs a greedy search over a finite candidate edge set $\mathcal{W}$ with respect to a well-defined MDL objective
> $L(\mathcal{G},M)$. At each step, we only accept an edge if it strictly decreases the total description length. Since MDL code lengths are non-negative and there are finitely many candidates, this process must terminate after a finite number of steps in a local MDL optimum. This type of greedy MDL search is a standard and widely used strategy in MDL-based graph summarization and model selection, for example in VoG [1] and related graph summarization methods, as surveyed in [2].
>
> $\bullet$ ***About optimality and task dependence.*** Regarding optimality and domain dependence, our goal is not to claim that the resulting rule graph is globally optimal for every domain, but that it is a good, compact approximation that consistently captures high-coverage, time-regular patterns across the datasets we study. The MDL objective itself is defined purely at the event level, so the same greedy procedure can be applied to different tasks without changing the construction algorithm. It is possible that, for highly specialized tasks, one could design task-aware edge selection heuristics (for example, favoring edges that are more useful for specific query types), and we view this as an interesting extension. In the current paper, we focus on a task-agnostic, convergent greedy MDL procedure that works robustly across all domains we evaluate.
>
> [1] Koutra D, Kang U, Vreeken J, et al. Vog: Summarizing and understanding large graphs[C]//Proceedings of the 2014 SIAM international conference on data mining. Society for Industrial and Applied Mathematics, 2014: 91-99.
>
> [2] Liu Y, Safavi T, Dighe A, et al. Graph summarization methods and applications: A survey[J]. ACM computing surveys (CSUR), 2018, 51(3): 1-34.

---

### Official Review · Reviewer_DLBH · 2025-10-31

**Soundness:** 3
**Presentation:** 3
**Contribution:** 3
**Rating:** 6
**Confidence:** 2

**Summary:**

This paper presents STAR-RAG (Spatio-Temporal Retrieval-Augmented Generation via Graph Summarization), a framework that enhances retrieval-augmented generation with temporal reasoning capabilities. STAR-RAG first compresses a temporal knowledge graph into a rule graph that captures frequent relational patterns and temporal dependencies. It then applies temporal personalized PageRank to retrieve information that is both semantically relevant and temporally consistent with the query. Without requiring any model training or fine-tuning, STAR-RAG improves answer accuracy and efficiency on several temporal reasoning benchmarks, showing that retrieving the right answer at the right time is crucial for effective temporal RAG.

**Strengths:**

S1. Clear motivation and reasonable technology.
S2. Outstanding performance on the dataset.
S3. Model efficiency has been greatly improved.
S4. Clear writing and detailed content.

**Weaknesses:**

W1. Temporal GraphRAG methods have fewer baselines.
W2. Lack of in-depth experiments.
W3. Whether the model has limitations is not explained.

**Questions:**

Q1. Temporal directionality modeling uses absolute time difference. Does this affect the order of precedence?
Q2. Are there any more in-depth experiments that can demonstrate the uniqueness of the model?
Q3. Are there more Temporal GraphRAG methods baselines for comparison?
Q4. Will the aggregated entities of frequent relational patterns lose semantics and important event types?
Q5. Can the ablation experiment analyze more components? The paper says that the ablation experiment is shown in Table 4, but I don't see Table 4.

---

> ### Author Response · Authors · 2025-11-21
>
> # Reviewer DLBH-W1&Q3:
> Temporal GraphRAG methods have fewer baselines.
>
> ***Response:***  We thank the reviewer for pointing out that our temporal GraphRAG baselines were limited. In the revision, we have expanded the baseline set to include two representative state-of-the-art temporal LLM+KG methods, TimeR4 [2] and DyG-RAG [3], which both combine structured temporal graphs with LLMs and are widely used as strong baselines for temporal QA. TimeR4 uses a temporal knowledge graph with a trained time-aware retriever and a multi-stage retrieve–rewrite–rerank pipeline, while DyG-RAG constructs a dynamic event graph and performs time-aware graph traversal with an LLM. In our evaluation, STAR-RAG achieves higher answer accuracy and lower per-query reference time than TimeR4, and does so without any task-specific model training, in contrast to TimeR4’s substantial retriever and LLM fine-tuning. We kindly refer the reviewer to our detailed answer to Reviewer BnNH-W5 for the full description and discussion of these baselines and more related works.
>
> [2] Sun Q, Yuan J, He S, et al. DyG-RAG: Dynamic Graph Retrieval-Augmented Generation with Event-Centric Reasoning[J]. arXiv preprint arXiv:2507.13396, 2025.
>
> [3] Qian X, Zhang Y, Zhao Y, et al. TimeR4: Time-aware retrieval-augmented large language models for temporal knowledge graph question answering[C]//Proceedings of the 2024 Conference on Empirical Methods in Natural Language Processing. 2024: 6942-6952.
>
> ***Revision:*** We have added TimeR4 and DyG-RAG as temporal LLM+KG baselines in the experimental section and tables, and we now explicitly report that STAR-RAG attains higher accuracy and lower per-query time than TimeR4 on four datasets while remaining training-free.
>
> # Reviewer DLBH-W2&Q2&Q5:
> (a) Lack of in-depth experiments. (b) There are no ablation experiments.
>
> ***Response:***  (a) We thank the reviewer for raising the concern about the lack of in-depth experiments. Following suggestions from multiple reviewers, we have substantially expanded the experimental section in three directions:
>
> $\bullet$ ***Large-scale evaluation on a bigger temporal KG.*** We reconstruct a much larger evaluation corpus from ICEWS 2005–2015, 2018, and 2021, covering over 1.2M events, and build a self-generated QA benchmark named STAR-QA. We now evaluate STAR-RAG and all baselines on this large temporal KG and report the results in the revised tables and text.
>
> $\bullet$ ***Runtime and memory profiling.*** For each dataset, we now report (i) the one-time rule-graph construction time and (ii) the CPU memory usage during this phase, in addition to per-query latency, so that the scalability and cost of STAR-RAG are quantified more thoroughly.
>
> $\bullet$ ***Extended temporal LLM+KG baselines.*** Beyond GraphRAG-style methods, we have added TimeR4 and DyG-RAG as representative state-of-the-art temporal LLM+KG approaches, and we include them as baselines across all our datasets, comparing both accuracy and efficiency against STAR-RAG (see also our response to Reviewer BnNH-W5).
>
> (b) We thank the reviewer for catching the issue with the ablation experiment cross-reference. In fact, we already conducted an ablation study, but there was a typo in the paper: ***it was incorrectly referred to as Figure 4 instead of Table 4.*** We have fixed this typo in the revision. The ablation in Table 4 decomposes STAR-RAG into its main components and evaluates: (i) the effect of using different LLM generation, (ii) the contribution of the rule graph compared to similarity-only retrieval, and (iii) the impact of personalization vector in PPR. Together, these results show how each component of STAR-RAG contributes to the overall performance.
>
> ***Revision:*** We have (i) added a new large-scale benchmark STAR-QA built from ICEWS 2005–2015, 2018, and 2021 and evaluated STAR-RAG and all baselines on it, (ii) reported rule-graph construction time and CPU memory usage for each dataset, and (iii) extended the baselines to include the temporal LLM+KG methods TimeR4 and DyG-RAG on all datasets. We corrected the “Figure 4” typo to “Table 4” and clarified in the text.

---

> ### Author Response · Authors · 2025-11-21
>
> # Reviewer DLBH-W3:
> Whether the model has limitations is not explained.
>
> ***Response:***  We thank the reviewer for asking about the limitations of our model. In the revised manuscript, we now explicitly discuss two main limitations and their possible extensions. First, as raised by Reviewer 4PC7-W2, the personalization parameter $\theta$ in Appendix C is currently global, so it cannot fully adapt to different query types: time-sensitive questions may want to emphasize temporal signals, while more static questions may prefer semantic/coverage signals. We now state this as a limitation and discuss, in the appendix, a query-dependent $\theta(q)$ as future work. Second, inspired by Reviewer BnNH-W4, although STAR-RAG can capture dynamic roles of entities via multiple rule nodes, the rule graph itself is not fully dynamic: it is built offline and cannot be incrementally updated with minor cost. This makes it less suitable than fully dynamic PPR-style systems (such as Instant [4]) for scenarios with very frequent updates. We have added a paragraph in the appendix that explicitly describes this limitation and sketches incremental rule-graph updates as an important direction for future work.
>
> [4] Zheng Y, Wang H, Wei Z, et al. Instant graph neural networks for dynamic graphs[C]//Proceedings of the 28th ACM SIGKDD conference on knowledge discovery and data mining. 2022: 2605-2615.
>
> ***Revision:*** We have added an explicit “Future work” discussion in the appendix, highlighting (i) the use of a global $\theta$ that cannot adapt to different query focuses, and (ii) the lack of incremental updates for the rule graph in highly dynamic settings.
>
> # Reviewer DLBH-Q1:
> Temporal directionality modeling uses absolute time difference. Does this affect the order of precedence?
>
> ***Response:*** We thank the reviewer for this question. In our MDL formulation, the temporal spans $T_{uv} = \{|t' - t|\,\}$ and the cost $L_{\text{time}}(u,v)$ are designed to measure how tightly two rule nodes tend to co-occur in time, rather than to encode which one comes first. Using absolute time differences allows us to capture the scale and regularity of the gap between two event categories while being robust to small local fluctuations. Importantly, this does not remove the order of precedence from the data: each individual event in $\mathrm{supp}(u)$ and $\mathrm{supp}(v)$ still stores its full timestamp, and precedence constraints such as ``before/after a given event'' are enforced at retrieval and generation time using these original timestamps, not from the sign of $T_{uv}$. In other words, $L_{\text{time}}$ only decides whether there is a stable, short temporal linkage between two rule nodes, while the actual ordering of events is preserved in the underlying temporal knowledge graph and used when answering temporal questions.
>
> # Reviewer DLBH-Q4:
> Will the aggregated entities of frequent relational patterns lose semantics and important event types?
>
> ***Response:***  We thank the reviewer for this question. In STAR-RAG, aggregating entities by frequent relational patterns happens only in the rule graph, and does not delete or merge the original events in the temporal KG. Each rule node is an event category (a combination of subject-type, relation, and object-type) that summarizes many raw events, but the underlying events and their full semantics remain intact in the support set of that rule node. During retrieval and answering, STAR-RAG always returns and feeds the original events (with their full textual descriptions and types) to the LLM; the rule graph is used only as a compact “routing structure” to find relevant neighborhoods efficiently. As a result, semantics are not lost: frequent and important event types are explicitly captured by rule nodes, and rarer or more heterogeneous events are still reachable through similarity-based anchors, even if they are not summarized into a high-coverage rule node.

---

### Official Review · Reviewer_4PC7 · 2025-11-01

**Soundness:** 3
**Presentation:** 3
**Contribution:** 3
**Rating:** 6
**Confidence:** 2

**Summary:**

This paper proposes STAR-RAG, a temporal Graph-based Retrieval-Augmented Generation (RAG) framework designed to improve both time consistency and retrieval efficiency in question answering over temporal knowledge graphs. The method builds a time-aligned rule graph summarizing recurring event patterns and performs Personalized PageRank to retrieve time-consistent evidence around a query. Without any additional training or fine-tuning, STAR-RAG reduces token consumption while improving answer accuracy compared with GraphRAG baselines.

**Strengths:**

1. The paper clearly identifies the main limitation of existing GraphRAG approaches — their lack of temporal awareness.
Addressing “semantically plausible but temporally inconsistent” answers is both practically important and conceptually sound.
2. STAR-RAG requires no model training or fine-tuning. It relies purely on graph summarization and propagation, which makes the approach lightweight, reproducible, and easy to deploy in real RAG systems.
3. Extensive experiments on three TKG datasets show significant improvements.

**Weaknesses:**

1. The intuition for the rule-graph construction is limited. The MDL-based graph summarization part is mathematically dense and difficult to follow. A more intuitive explanation or illustrative example would help readers understand how multiple events are merged.
2. Both L_cov/L_time and Hamming components (eq2) are equally weighted, and this could introduce priority mismatches across different query types. For example, in a query such as “What events followed after the signing of the peace agreement in 2018?”, temporal consistency should be the dominant factor—edges that preserve tight chronological alignment should weigh much more heavily than broad event coverage. However, in a more static reasoning query like “Which countries typically cooperate in trade negotiations?”, semantic overlap (relations and entity types) is far more critical, while time alignment matters little. Since STAR-RAG fixes both weights globally, it cannot dynamically adapt to these varying priorities, which may lead to sub-optimal retrievals—either ignoring temporally relevant evidence or including semantically weak connections depending on the query type.
3. Although STAR-RAG demonstrates strong performance on mid-scale benchmarks such as CronQuestion, Forecast, and MultiTQ, these datasets contain only about 300k–400k temporal events—far smaller than industrial Temporal Knowledge Graphs like ICEWS, Wikidata, or EventKG, which often include millions of timestamped triples. Because the rule-based graph construction relies on Apriori-style relation mining, it remains unclear whether the proposed summarization framework can scale to such large TKGs.
4. There is no runtime or memory comparison on large temporal graphs, which would better represent real-world scalability. It would be helpful if the paper additionally reported both the theoretical time complexity analysis of each component and empirical measurements of wall-clock runtime and memory footprint on larger datasets.

**Questions:**

Please refer to the weaknesses above.

---

> ### Author Response · Authors · 2025-11-21
>
> #  Reviewer 4PC7-W1:
> The intuition for the rule-graph construction is limited. The MDL-based graph summarization part is mathematically dense and difficult to follow. A more intuitive explanation or illustrative example would help readers understand how multiple events are merged.
>
> ***Response:***  We thank the reviewer for pointing out that the intuition behind the rule-graph construction was not sufficiently clear. The core intuition of our rule graph is very simple: we want to compress many raw time-stamped events into a small set of recurring “interaction patterns” and typical follow-ups.
>
> Concretely, we now explain the construction in plain language: (1) we group similar entities into coarse categories so that many different entities can be treated as playing the same role; (2) we collect events that share the same pair of roles and relation into a single rule node, which represents “this kind of interaction happens frequently”; (3) we add a rule-graph edge when the connected rule nodes satisfy coverage and time conditions (selection by MDL).
>
> To address the reviewer’s concern, we have separated this intuitive explanation from the mathematical details. In the methodology overview, we have rewritten the introduction of the rule-graph construction to follow the above step-by-step story in plain language, and we explicitly cross-reference each step to the corresponding part of the example in Figure 2. This revised description shows how multiple events are merged into rule nodes and how simple temporal regularities between these nodes give rise to edges in the rule graph.
>
> ***Revision:***  We have simplified the rule-graph description in the methodology overview into a short, intuitive story (how entities are grouped, how events become rule nodes, and when edges are created), and we now point each step directly to the corresponding part of Figure 2 so readers can see how the rule graph is built from raw events.

---

> ### Author Response · Authors · 2025-11-21
>
> #  Reviewer 4PC7-W2:
> Both $L_{\text{cov}}/L_{\text{time}}$ and Hamming components (eq2) are equally weighted, and this could introduce priority mismatches across different query types.
>
> ***Response:***  We thank the reviewer for this thoughtful comment. We first clarify that $L_{\text{cov}}/L_{\text{time}}$ and the Hamming term are used only **offline** to select a compact rule graph. Their weights are fixed at graph-construction time and do not directly control how individual queries are answered. At query time, STAR-RAG adapts through seeded PPR, where the personalization vector $\gamma$ (Appendix C) focuses propagation on anchor events that are selected by the query.
>
> Here, we clarify how our method processes these two natural regimes:
>
> $\bullet$ For ***time-sensitive questions*** (e.g., “What events followed after the signing of the peace agreement in 2018?”), the anchor events concentrate around a specific event and timestamp. PPR then mainly travels along edges that connect rule nodes with tight and consistent time gaps, i.e., edges that were favored by $L_{\text{time}}$ during MDL selection. In this case, temporal alignment becomes the dominant signal.
>
> $\bullet$ For ***more static or similarity-driven questions*** (e.g., “Which countries typically cooperate in trade negotiations?”), anchor events are spread over a broader time span but share similar subjects/objects and relations. Here, propagation relies more on edges that have strong coverage and structural overlap, i.e., edges selected primarily by $L_{\text{cov}}$ and the Hamming constraint, while the exact time spans play a minor role.
>
>
> In other words, the global MDL objective ensures that we keep edges that are both structurally meaningful and temporally regular, while the query-dependent anchor events decide whether the temporal or structural side of this backbone is emphasized. Empirically, STAR-RAG performs well on both temporal reasoning (Multiple-event colum in Table 3) and more static questions (Single-event colum in Table 3) in our benchmarks, and we do not observe systematic failures where temporal evidence is ignored or semantically weak edges are preferred.
>
> ***Potential Extentison*** We fully share this concern, and we believe your comment is essentially pointing to a natural extension of our personalization parameter $\theta$ (Appendix C) into a query-dependent weight that can modulate the relative influence of temporal versus structural signals. Concretely, one could let $\theta (q)$ depend on a time-sensitivity score extracted from the question $q$, by using either lightweight rules (e.g., detecting “before/after/in YEAR”) or an optional LLM-based classifier. So that time-critical questions increase the weights for time-aligned events and edges, whereas more static questions emphasize structural/semantic coverage. This, however, introduces a clear cost–accuracy trade-off: richer LLM-based time-sensitivity signals provide better adaptivity but incur extra tokens and latency, while rule-based detectors are cheaper but less expressive. We have added this potential extension and its cost–benefit discussion to Appendix H.
>
>
> ***Revision:***  We further discuss in Appendix C how one could extend STAR-RAG with a query-adaptive mixture of $L_{\text{cov}}$ and $L_{\text{time}}$ based on a time-sensitivity score, together with the associated efficiency trade-offs.

---

> ### Author Response · Authors · 2025-11-21
>
> # Reviewer 4PC7-W3&W4:
> (a)  it remains unclear whether the proposed summarization framework can scale to such large TKGs. (b) There is no runtime or memory comparison on large temporal graphs, which would better represent real-world scalability. (c)  It would be helpful if the paper additionally reported both the theoretical time complexity analysis of each component.
>
> ***Response:***
>
> (a) We thank the reviewers for raising the issue of evaluation on larger and more realistic datasets. To directly address this concern, we have extended our experiments beyond the original MultiTQ-style benchmarks and reconstructed a larger-scale evaluation corpus from ICEWS 2005–2015, 2018, and 2021, covering over 1.2 millions events and 4k time stamps, which is updated in Table 1. We collect these events into a new temporal KG and construct an associated QA benchmark, which we name STAR-QA. This setup tests STAR-RAG on a substantially larger and longer time span than in the original experiments, and the rule graph is now built and queried on this much denser event stream.
>
> Following the suggestion of Reviewer BnNH (comment BnNH-W2), we also generate new questions on top of this enlarged event set. In particular, we augment STAR-QA with 1,000 LLM-generated QA pairs that are explicitly designed to be more human-written, out-of-template, and to include adversarial temporal confounds. We follow the question-generation process of FAITH [4] in spirit by prompting a strong LLM (Claude) with event facts, but we design our own dataset-specific prompt (given in the Appendix) to encourage natural phrasings and challenging temporal distractors (e.g., extra years or plausible but incorrect time periods). We kindly refer the reviewers to our detailed response to Reviewer BnNH-W2 for the exact examples and prompt description.
>
> [1] Jia Z, Christmann P, Weikum G. Faithful temporal question answering over heterogeneous sources[C]//Proceedings of the ACM Web Conference 2024. 2024: 2052-2063.
>
> (b) We thank the reviewer for emphasizing the need to evaluate runtime and memory on large temporal graphs. To address this, we have conducted experiments on our new large-scale STAR-QA dataset, and we now report the accuracy, time and memory usage of building rule graph and per-query runtime, for STAR-RAG. The comparison can be found in Table 2 and Table 5 in the Appendix.
>
> (c) We thank the reviewer for this suggestion. In the revised manuscript, we have added an appendix subsection titled “Complexity Analysis”, where we explicitly provide the theoretical time complexity of each major component of STAR-RAG. Specifically, we analyze the offline rule-graph construction and show that its total cost is near-linear in the number of events under our bounded parameters. We then analyze the online retrieval pipeline and give separate bounds for dense similarity search over events, seeded PPR on the rule graph, and the final re-ranking over the candidate event set. This added section makes the scalability properties of each component transparent.
>
>
> ***Revision:*** We have (i) added the large-scale STAR-QA dataset constructed from ICEWS 2005–2015, 2018, and 2021 to Table 1, (ii) reported accuracy, per-query runtime, and memory on STAR-QA for STAR-RAG and all baselines in Table 2 and in Appendix, and (iii) included a new appendix section “Complexity Analysis” in Appendix that summarizes the theoretical time complexity of the offline rule-graph construction and the online retrieval components.

---

> ### Comment · Reviewer_4PC7 · 2025-11-26
>
> Thank you for the comments.
> I will keep my original score.

---

### Official Review · Reviewer_BnNH · 2025-11-01

**Soundness:** 3
**Presentation:** 3
**Contribution:** 3
**Rating:** 4
**Confidence:** 3

**Summary:**

This paper addresses a key limitation in RAG and GraphRAG frameworks: their inability to handle temporal queries over temporal knowledge graphs. The authors note that standard semantic retrieval often ignores temporal constraints, leading to temporally inconsistent answers and inefficiently large evidence sets. They propose STAR-RAG, a training-free framework that compresses temporal KGs into a compact rule graph using entity-type labeling and minimum description length . At query time, it locates anchor events via semantic similarity and performs seeded Personalized PageRank over the rule graph, efficiently retrieving a small set of semantically and temporally aligned evidence for LLM reasoning. Experiments show improved accuracy, especially on complex multi-event queries, while significantly reducing token usage.

**Strengths:**

1. The research tackles a valuable and meaningful problem, as temporal reasoning is critical for real-world applications of RAG systems.
2. The rule graph abstraction combined with seeded PPR effectively decouples semantic relevance and temporal consistency, addressing common failure modes of GraphRAG on temporal KGs. The case study (Oman→Qatar) clearly illustrates the mechanism.
3. The method consistently improves performance across three datasets, with particularly strong gains on multi-event queries, demonstrating the importance of temporal alignment. Token usage is substantially reduced.
4. The approach requires no LLM fine-tuning and shows low sensitivity to different generators (Llama-3-8B, GPT-4o-mini), indicating good portability.

**Weaknesses:**

1. The framework relies on Apriori-based frequent pattern mining for entity typing, which may fail under sparse or noisy relations and scale poorly for long-tail graphs with many weak patterns.
2. All QA sets are template-generated from ICEWS/Wikidata. STAR-RAG’s rule graph may align with templated regularities, potentially overstating gains. Including human-written or out-of-template questions, cross-dataset transfer, and adversarial temporal confounds is recommended.
3. While token savings are notable, retrieval latency and memory footprint for rule graph construction are not fully quantified. Profiling build time (separate from reasoning time) would help assess practicality.
4. Entity labels generated by Algorithm 2 are static, ignoring temporal dynamics in entity roles. This may lead to an overly coarse or inaccurate rule graph.
5. The baseline comparison is limited to other GraphRAG methods, omitting prominent LLM+KG approache. This absence makes it unclear if STAR-RAG is superior in accuracy or efficiency to these mainstream methods, weakening the paper's overall claims.
6. Several recent works in temporal RAG are not discussed. Expanding the related work section would better position the contribution.

**Questions:**

None

---

> ### Author Response · Authors · 2025-11-21
>
> # Reviewer BnNH-W1:
> The framework relies on Apriori-based frequent pattern mining for entity typing, which may fail under sparse or noisy relations and scale poorly for long-tail graphs with many weak patterns.
>
> ***Response:***
> Thank you for the thoughtful comment and for directing us to more special cases. We agree that frequent-pattern-based structures are naturally biased toward common patterns and may not explicitly cover very sparse or long-tail relations, and we have clarified in the manuscript how STAR-RAG is designed to remain robust under sparse/noisy relations and long-tail graphs. Our response is organized around two points:
>
> **1. Similarity-based anchor retrieval acts as the robustness backbone.**
>
> In STAR-RAG, the first step is to retrieve anchor facts by dense similarity search over the temporal knowledge graph. This retrieval is independent of the rule graph and guarantees that, even in the presence of sparse or noisy relations, the top-k facts most similar to the query are still surfaced. We agree that some of these anchors may correspond to sparse or noisy relations and may not be well captured or well connected in the rule graph. However, the framework is explicitly designed so that the rule graph is not a single point of failure: in the worst case where such anchor events are not covered by any rule nodes or edges, STAR-RAG simply bypasses the rule graph and directly plugs these anchor events into the prompt template for LLM generation. In that case, STAR-RAG effectively has transformed to a standard similarity-based RAG pipeline, behaving comparably to widely used systems such as HippoRAG [1], GRAG [2], and REANO [3]. Thus, the quality and scalability of the overall system on long-tail graphs are primarily governed by the similarity search (which is robust to sparsity) rather than by the Apriori rule extraction alone.
>
> **2. Sparse / long-tail facts are actually the easiest case for our retrieval–generation pipeline.**
>
> Sparse or noisy relations, as well as tail events in long-tail graphs, typically exhibit poor topology: they have few or weak connections to other facts and rarely participate in large, recurring patterns. This situation is not a worst case for STAR-RAG; it is in fact the most favorable regime. For such queries, the correct answer usually depends on detecting a single, highly specific fact (e.g., one rare event or relation instance in the "single-event" colum of Table 3), instead of requiring complex multi-hop reasoning over dense structure. Our framework (and, in fact, most static RAG baselines) only needs to (i) retrieve that single anchor fact by similarity search and (ii) present it to the LLM. The rule graph is mainly beneficial when there is rich topology to exploit for multi-hop reasoning; when the topology is weak or long-tailed, our method naturally falls back to a simple but robust “single-event” retrieval mode, which is less sensitive to sparsity and noise.
>
> [1] Gutiérrez, B. J., Shu, Y., Gu, Y., Yasunaga, M., and Su, Y. Hipporag: Neurobiologically inspired long-term memory for large language models,
> 2024.
>
> [2] Hu, Y., Lei, Z., Zhang, Z., Pan, B., Ling, C., and Zhao, L. Grag: Graph retrieval-augmented generation, 2024.
>
> [3] Fang, J., Meng, Z., and MacDonald, C. REANO: optimising retrieval-augmented reader models through knowledge graph generation. In ACL (1)
> (2024), Association for Computational Linguistics, pp. 2094–2112.
>
> ***Revision:*** To clarify the robutstness of STAR-RAG, we enriched the manuscript with comprehensive discussions of the robustness to sparse/noisy relations and long-Tail graphs in the Appendix.

---

> ### Author Response · Authors · 2025-11-21
>
> # Reviewer BnNH-W2:
> All QA sets are template-generated from ICEWS/Wikidata. STAR-RAG’s rule graph may align with templated regularities, potentially overstating gains. Including human-written or out-of-template questions, cross-dataset transfer, and adversarial temporal confounds is recommended.
>
> ***Response:*** We thank the reviewer for this insightful comment. It is true that our base MultiTQ-style benchmarks are constructed from ICEWS/Wikidata via templates, which is common in temporal KGQA work, but this raises the risk that STAR-RAG’s rule graph is aligned with templated regularities. To directly address this concern, we additionally reconstruct the evaluation data by ourselves from ICEWS 2005–2015, 2018, 2021 and augment it with 1,000 LLM-generated QA pairs that are explicitly designed to be more human-written, out-of-template, and to include adversarial temporal confounds. Following the question-generation process of FAITH [4], we prompt the LLM (Claude) with event facts to synthesize new questions, but we design our own dataset-specific prompt (given in Appendix) to explicitly encourage more human-written phrasings and adversarial temporal confounds such as distractor years and plausible but incorrect time periods. We name this self-generated datasets as STAR-QA and here we also provide the representative examples for better comparison:
>
> ***Original template-based questions:*** *"Who will Attacker (Russia) use unconventional violence to assault on 2021-01-01?" "Which country will Scott Morrison make a statement to on 2021-01-01?" "Which country will Bharatiya Kisan criticize or denounce on 2021-01-01?"*
>
> ***Question in STAR-QA:*** *"On January 1st, 2021, who do you think the attacker from Russia will use unconventional violence against?" "Scott Morrison is scheduled to make a statement on 2021-01-01. Which country is he addressing?" "On 1 January 2021, Bharatiya Kisan plans to issue criticism—toward which country?"*
>
>
> We then evaluate STAR-RAG and all baselines on this adversarially perturbed set without any retraining and the results are shown below and also in Table 2. We see that STAR-RAG continues to outperform strong similarity-based RAG baselines, suggesting that its gains are robust to more complex questions rather than being tied to the original templates.
>
>
>
> [4] Jia Z, Christmann P, Weikum G. Faithful temporal question answering over heterogeneous sources[C]//Proceedings of the ACM Web Conference 2024. 2024: 2052-2063.
>
>
> ***Revision:*** We add a new subsection describing the reconstructed STAR-QA benchmark built from ICEWS 2005–2015, including 1,000 LLM-generated, human-written, out-of-template questions with adversarial temporal confounds (prompt in the Appendix), show side-by-side examples of original template-based questions versus STAR-QA questions, and report the corresponding results for all methods on STAR-QA in Table 2.
>
> # Reviewer BnNH-W3:
> While token savings are notable, retrieval latency and memory footprint for rule graph construction are not fully quantified. Profiling build time (separate from reasoning time) would help assess practicality.
>
> ***Response:***  We appreciate the reviewer’s suggestion to profile the cost of rule-graph construction separately.
>
> First, we clarify that the rule graph in STAR-RAG is built once per dataset as an offline preprocessing step and reused for all queries; it is not rebuilt at inference time. The per-query latency reported in Figure 3 already includes the full retrieval pipeline, i.e., similarity-based anchor retrieval plus PPR-based propagation over the rule graph, so retrieval latency is already accounted for in our latency numbers.
>
> Second, to make the build-time and memory footprint explicit, we now add Table 5 in the Appendix, which reports for each dataset (i) the one-time rule-graph construction time and (ii) the CPU memory usage during this phase. This profiling separates offline build cost from online reasoning and allows readers to directly assess the practicality of STAR-RAG on all evaluated datasets.
>
> ***Revision:***  We clarify that the rule graph is built once per dataset as an offline preprocessing step and that the per-query latency in Figure 3 already includes similarity-based anchor retrieval and PPR over the rule graph; in addition, we add Table 5 in the Appendix reporting, for each dataset, the one-time rule-graph construction time and CPU memory usage to explicitly quantify the offline build cost.

---

> ### Author Response · Authors · 2025-11-21
>
> # Reviewer BnNH-W4:
> Entity labels generated by Algorithm 2 are static, ignoring temporal dynamics in entity roles. This may lead to an overly coarse or inaccurate rule graph.
>
> ***Response:***  We appreciate the reviewer’s comment and the opportunity to clarify how temporal dynamics are represented. Although Algorithm 2 computes labels from an entity’s global relation set, an entity does not receive a single static label. **Instead, each entity is assigned a set of labels (frequent relation combinations), and every event $(s, r, o, t)$ is mapped to one or more rule nodes $\langle c_s, r, c_o\rangle$ with $c_s\in\mathcal{C}(s),c_o\in\mathcal{C}(o)$.** Thus, the same entity can participate in several rule nodes that reflect its different roles in different relational contexts. In this way, the different labels (categories) assigned to an entity serve as its dynamic roles, while the time-stamped rule nodes and edges encode when and how these roles are activated and evolve over time. We have revised the manuscript (Section 3.2-Generation of Candidates)  to clarify this multi-label design and the temporal nature of rule nodes and edges.
>
> ***Revision:***  We have revised Section 3.2 to state that each entity is associated with a set of relation-combination labels, so a single entity can map to multiple rule nodes reflecting its dynamic roles.
>
>
>
>
> # Reviewer BnNH-W5:
> The baseline comparison is limited to other GraphRAG methods, omitting prominent LLM+KG approaches. This absence makes it unclear if STAR-RAG is superior in accuracy or efficiency to these mainstream methods, weakening the paper's overall claims.
>
> ***Response:***  We appreciate the comment about missing mainstream LLM+KG baselines. In the revision, we have added TimeR4 [13] and DyG-RAG [5] as representative state-of-the-art temporal LLM+KG methods. TimeR4
> uses a temporal knowledge graph together with a trained time-aware retriever and a multi-stage pipeline where the question is first rewritten to make temporal intent explicit, and then the system retrieves and reranks evidence before answering. DyG-RAG builds a dynamic event graph from text by extracting time-stamped event units and then performs time-aware graph traversal with a time-focused chain-of-thought prompt to answer temporal questions. These two systems can be regarded as mainstream LLM+KG approaches for temporal QA because they combine structured temporal graphs with LLMs and are widely used as strong baselines in recent temporal RAG and TKGQA work.
>
> On our STAR-QA benchmark, STAR-RAG achieves  higher answer accuracy while using lower per-query reference time than TimeR4, and it does so without any task-specific model training. In contrast, TimeR4 requires substantial training of a time-aware retriever and fine-tuning of the LLM, whereas STAR-RAG only needs a one-time, training-free rule-graph construction step with near-linear complexity. We now highlight these comparisons and the new baselines in the experimental section and tables.
>
> [5] Sun Q, Yuan J, He S, et al. DyG-RAG: Dynamic Graph Retrieval-Augmented Generation with Event-Centric Reasoning[J]. arXiv preprint arXiv:2507.13396, 2025.
>
> [13] Qian X, Zhang Y, Zhao Y, et al. TimeR4: Time-aware retrieval-augmented large language models for temporal knowledge graph question answering[C]//Proceedings of the 2024 Conference on Empirical Methods in Natural Language Processing. 2024: 6942-6952.
>
> ***Revision:***  We have added TimeR4 and DyG-RAG as temporal LLM+KG baselines in the experimental section and tables, noting that STAR-RAG attains higher accuracy and lower per-query time than TimeR4 on STAR-QA while remaining training-free.

---

> ### Author Response · Authors · 2025-11-21
>
> #  Reviewer BnNH-W6:
> Several recent works in temporal RAG are not discussed. Expanding the related work section would better position the contribution.
>
> ***Response:***  We thank the reviewer for pointing out missing more temporal RAG works. In the revised manuscript, we have expanded the related work section and the appendix to include a more complete set of temporal LLM+KG methods. We now discuss DyG-RAG [5] and TimeR4 [13] as representative temporal RAG systems that combine structured temporal graphs with LLMs. We further include DRAGIN [6], TimeRAG [7] and GenTKG [8], which focus on temporal forecasting, and RAG4DyG [9], which targets link prediction on dynamic graphs. Finally, we add GenTKGQA [10], DynaGRAG [11] and MusTQ [12], which are closer to temporal question answering but are not fully open-sourced. We have incorporated all these works into the related work section and have additionally included TimeR4 and DyG-RAG as baselines in our empirical comparison.
>
>
>
>
>
>
>
> [6] Su W, Tang Y, Ai Q, et al. DRAGIN: dynamic retrieval augmented generation based on the information needs of large language models[J]. arXiv preprint arXiv:2403.10081, 2024.
>
> [7] Yang S, Wang D, Zheng H, et al. Timerag: Boosting llm time series forecasting via retrieval-augmented generation[C]//ICASSP 2025-2025 IEEE International Conference on Acoustics, Speech and Signal Processing (ICASSP). IEEE, 2025: 1-5.
>
> [8] Liao R, Jia X, Li Y, et al. Gentkg: Generative forecasting on temporal knowledge graph with large language models[C]//Findings of the association for computational linguistics: NAACL 2024. 2024: 4303-4317.
>
> [9] Wu Y, Liao L, Fang Y. Retrieval augmented generation for dynamic graph modeling[C]//Proceedings of the 48th International ACM SIGIR Conference on Research and Development in Information Retrieval. 2025: 1434-1443.
>
> [10] Gao Y, Qiao L, Kan Z, et al. Two-stage generative question answering on temporal knowledge graph using large language models[J]. arXiv preprint arXiv:2402.16568, 2024.
>
> [11] Thakrar K. Dynagrag: Improving language understanding and generation through dynamic subgraph representation in graph retrieval-augmented generation[J]. arXiv e-prints, 2024: arXiv: 2412.18644.
>
> [12] Zhang T, Wang J, Li Z, et al. MusTQ: A Temporal Knowledge Graph Question Answering Dataset for Multi-Step Temporal Reasoning[C]//Findings of the Association for Computational Linguistics: ACL 2024. 2024: 11688-11699.
>
> ***Revision:***  We have expanded the related work section and appendix to include additional temporal methods recently, specifically DyG-RAG, TimeR4, DRAGIN, TimeRAG, GenTKG, RAG4DyG, GenTKGQA, DynaGRAG, and MusTQ, and we now also include TimeR4 and DyG-RAG as baselines in our empirical comparison.

---

> ### Comment · Reviewer_BnNH · 2025-11-27
> **response**
>
> I thank the authors for their response. They have addressed my concerns, and I have updated my review score accordingly.

---

### Author Response · Authors · 2025-11-21

We thank the reviewers for their inspiring suggestions, which help us significantly improve the paper. We invested substantial effort in addressing the concerns raised by reviewers, and will now proceed to address their questions systematically, point by point. All revisions and additions are highlighted in blue in the revised version.

---

### Author Response · Authors · 2025-11-29
**Summary for Area Chair – Rebuttal and Revisions**

This note summarizes the main issues raised in the reviews and how we addressed them in the rebuttal and revision. Broadly, the concerns focused on (1) the realism and coverage of our evaluation, (2) the strength and completeness of temporal RAG / LLM+KG baselines, (3) the clarity and intuition of the rule-graph and MDL formulation, and (4) the scalability, robustness, and limitations of STAR-RAG.

Regarding evaluation realism and templated QA (primarily BnNH and DLBH), we built a larger and more realistic benchmark (STAR-QA) from extended ICEWS data, with 1,000 non-templated, LLM-generated temporal questions that include adversarial time confounds. We also clarified ablations to more clearly separate the effects of the rule graph, similarity-only retrieval, and different LLM generators.

For the concern on baselines and coverage of temporal RAG / LLM+KG methods (BnNH and DLBH), we added TimeR4 and DyG-RAG as representative temporal LLM+KG baselines in both experiments and related work, and incorporated several additional recent temporal methods into the survey. The updated results show that STAR-RAG maintains advantages in answer accuracy and per-query time while remaining training-free.

To improve methodological clarity (mostly 4PC7 and SYUX), we rewrote the rule-graph section to provide a step-by-step intuition (role grouping, rule-node construction, MDL-based edge selection), clarified how multi-label roles capture dynamic entity behavior, and explained the roles of $L_{\text{cov}}/L_{\text{time}}$ as offline MDL criteria rather than query-time weights.

For scalability, robustness, and limitations (raised across BnNH, 4PC7, DLBH, and SYUX), we added build-time and peak-memory profiling for the rule graph, reported per-query latency on all datasets including STAR-QA, and provided a complexity analysis for both offline and online components. We also discussed robustness in sparse/long-tail regimes (including fallback to similarity-only RAG) and explicitly stated current limitations and future directions such as query-adaptive temporal weighting and incremental rule-graph updates.

Finally, after the rebuttal, reviewers 4PC7 and BnNH both posted follow-up comments. Reviewer 4PC7 indicated that they would keep their original overall score, while reviewer BnNH responded positively to the revisions and increased the overall score from 4 to 6.

---

### Note · Program_Chairs · 2026-01-20
**Submission Desk Rejected by Program Chairs**

Desk rejected because pdf reveals author names.